# Ubiquinol-mediated suppression of mitochondria-associated ferroptosis is a targetable function of lactate dehydrogenase B in cancer

Haibin Deng [1,2,3,4], Liang Zhao [3,4,5], Huixiang Ge[3,4,5], Yanyun Gao[3,4], Yan Fu [3,4,5], Yantang Lin[3,4,5], Mojgan Masoodi [6], Tereza Losmanova[7], Michaela Medová [2,8], Julien Ott[8], Min Su[1,2], Wenxiang Wang[1,2], Ren-Wang Peng [3,4] ✉, Patrick Dorn [3,4] ✉ & Thomas Michael Marti [3,4] ✉

Lactate dehydrogenase B (LDHB) fuels oxidative cancer cell metabolism by converting lactate to pyruvate. This study uncovers LDHB's role in countering mitochondria-associated ferroptosis independently of lactate's function as a carbon source. LDHB silencing alters mitochondrial morphology, causes lipid peroxidation, and reduces cancer cell viability, which is potentiated by the ferroptosis inducer RSL3. Unlike LDHA, LDHB acts in parallel with glutathione peroxidase 4 (GPX4) and dihydroorotate dehydrogenase (DHODH) to suppress mitochondria-associated ferroptosis by decreasing the ubiquinone (coenzyme Q, CoQ) to ubiquinol (CoQH2) ratio. Indeed, supplementation with mitoCoQH2 (mitochondria-targeted analogue of CoQH2) suppresses mitochondrial lipid peroxidation and cell death after combined LDHB silencing and RSL3 treatment, consistent with the presence of LDHB in the cell fraction containing the mitochondrial inner membrane. Addressing the underlying molecular mechanism, an in vitro NADH consumption assay with purified human LDHB reveals that LDHB catalyzes the transfer of reducing equivalents from NADH to CoQ and that the efficiency of this reaction increases by the addition of lactate. Finally, radiation therapy induces mitochondrial lipid peroxidation and reduces tumor growth, which is further enhanced when combined with LDHB silencing. Thus, LDHB-mediated lactate oxidation drives the CoQ-dependent suppression of mitochondria-associated ferroptosis, a promising target for combination therapies.

In his pioneering study, Warburg reported 100 years ago (1923) that the conversion rate of glucose to lactate is 70 times higher in tumor tissue compared to normal tissue[1]. Although initially considered a waste product of aerobic glycolysis (e.g., the Warburg effect), lactate has been shown to serve as the primary carbon source for the TCA cycle in vivo, providing substrates, and thus electrons, for oxidative phosphorylation in normal tissue and tumors[2,3].

The genes lactate dehydrogenase A and B (*LDHA* and *LDHB*, respectively) encode the tetrameric enzyme lactate dehydrogenase (LDH), which catalyzes the interconversion of pyruvate and lactate

using NADH/NAD+ as a co-substrate (reviewed in ref. 4). The activity of LDHA, particularly the LDHA homo-tetramer, is associated with converting pyruvate to lactate and is thus associated with the Warburg effect[5]. Historically, LDHB activity has been studied mainly in non-transformed cells in the context of the Cori cycle[6,7]. More recently, LDHB has been shown to be essential for the survival of cancer cells from various tissues of origin[8–10]. In this context, we have recently reported that silencing LDHB reduces lung cancer tumor growth and tumorigenesis[11]. On the molecular level, LDHB silencing induced persistent mitochondrial DNA damage, associated with decreased mitochondrial respiratory complex activity and oxidative phosphorylation, resulting in reduced levels of mitochondria-related metabolites. We found that silencing of LDHB in lung cancer cells resulted in only modest induction of apoptosis, which thus could not fully account for the significant loss of viability, sphere, and colony formation[11].

Ferroptosis is an iron-dependent, regulated cell death program independent of apoptosis associated with oxidative damage to phospholipids on cellular membranes[12,13]. Increased membrane lipid peroxidation depends on polyunsaturated fatty acid-containing phospholipid (PUFA-PL) synthesis, iron metabolism, and mitochondrial metabolism[12]. Although it does not directly assess lipid peroxidation products, the fluorescence assay using the organoboron C11-BODIPY581/591 (C11-BODIPY) conjugated to a fluorophore is a popular measure to assess the capacity of cells to oxidize the probe by a putative catalytic peroxidation mechanism[14,15]. Further, free radical attack on PUFA-PL can ultimately lead to multiple aldehydes with different carbon chain lengths, including 4-hydroxy-2-nonenal (4-HNE), an alpha, beta-unsaturated aldehyde, which is one of the major products of lipid peroxidation. 4-HNE forms stable covalent adducts to proteins, which can be detected by immunostaining with specific antibodies[15,16].

Continuous lipid peroxidation is counterbalanced by at least four defense systems with different subcellular localizations and molecular mechanisms[12]. Most prominently, glutathione (GSH), built from cysteine, glutamate, and glycine, serves as a co-factor and reducing substrate for glutathione peroxidase 4 (GPX4), which detoxifies lipid peroxides in both the cytoplasm and mitochondria and can be covalently bound and thus inactivated by RSL3[17]. Independent of GPX4, ferroptosis suppressor protein 1 (FSP1) operates in the plasma membrane and other non-mitochondrial membranes as a NAD(P)H-dependent oxidoreductase capable of reducing coenzyme Q, i.e., ubiquinone (CoQ10, hereafter CoQ) to ubiquinol (CoQH2), which serves as a trap for lipid peroxyl radicals. However, the most crucial function of CoQ is to serve as the major electron-transporting lipid of the mitochondrial electron transport system (ETS)[18]. In addition, CoQ serves as the electron acceptor for nine mitochondrial inner membrane dehydrogenases[18]. Recently, it was shown that the reduction of CoQ by dihydroorotate dehydrogenase (DHODH) in the mitochondrial membrane can compensate for the loss of GPX4 to detoxify mitochondrial lipid peroxidation[19]. A subsequent study showed that glycerol-3-phosphate is oxidized by G3P dehydrogenase 2 (GPD2) at the inner mitochondrial membrane, which also results in the reduction of CoQ, giving rise to CoQH2, and thus suppression of ferroptosis in mitochondria[20]. Interestingly, the function of the TCA cycle and the ETS are required for ferroptosis induced by cellular cysteine deprivation[21]. This indicates a delicate balance between the anti-ferroptotic function of the ETS, e.g., mitochondrial CoQH2 production, and the pro-ferroptotic functions, e.g., the production of ROS and ATP. In addition, the potential roles of additional mitochondrial enzymes contributing to the generation of CoQH2 remain to be elucidated.

We reported previously that silencing LDHB in *KRAS*-mutant lung cancer cell lines significantly reduced the expression of SLC7A11[11], which is required for oncogenic RAS transformation and encodes a plasma membrane-localized cystine/glutamate antiporter that provides cysteine for GSH synthesis[22,23]. Further, we described that LDHB protects specifically KRAS-mutant lung cancer cells from ferroptosis, mainly of the plasma membrane, by maintaining SLC7A11-mediated glutathione metabolism[24]. However, the SLC7A11-related function of LDHB cannot fully account for the mitochondrial phenotype described in our previous publication[11] and its mitochondrial localization described by others[25,26].

Here we show that LDHB protects cancer cells from mitochondria-associated ferroptosis via CoQ-dependent lactate oxidation, independently of lactate's role as a carbon source. Targeting LDHB enhances radiotherapy efficacy, highlighting its potential as a therapeutic target.

## Results

### Silencing of LDHB induces mitochondrial lipid peroxidation in cancer cells

Reanalysis of the gene expression data from our previous study revealed that silencing of LDHB in the NSCLC cell lines A549 and H358 was associated with a significant increase in gene expression of prostaglandin-endoperoxide synthase 2 (*PTGS2*), a biomarker of ferroptosis[17,19] (Supplementary Fig. 1a). Indeed, the flow cytometry-based measurement of the fluorescence level of intracellular oxidized C11-BODIPY revealed that LDHB silencing significantly increased total cellular lipid peroxidation in A549 cells and also in H460 cells, a lung large cell carcinoma cell line (Supplementary Fig. 1b–d). Previous studies, including ours, have shown that LDHB is localized in the mitochondria of cancer cells of different tissues of origin[11,25]. This led us to speculate whether LDHB is related to mitochondrial lipid peroxidation. Indeed, LDHB silencing resulted in a significant increase in mitochondrial lipid peroxidation not only in the NSCLC cell line A549, the malignant pleura mesothelioma (MPM) cell line MSTO-211H, the fibrosarcoma cell line HT1080, and the pancreatic ductal carcinoma cell lines PANC-1 (Fig. 1a–d) but in all thirteen tested cancer cell lines derived from different tissues of origin, e.g., the additional NSCLC cell lines H460, H358, H226, H1650, H2009, DMS114, and the primary NSCLC culture PF139, the additional MPM cell line H2452, the pancreatic ductal carcinoma cell line SU86.86, and the breast adenocarcinoma cell line CAL-85-1 (Supplementary Fig. 1e–h). Our flow cytometry-based results were confirmed by immunostaining with an antibody recognizing adducts of the lipid peroxidation product 4-HNE[27]. In detail, the level of 4-HNE staining were significantly increased by LDHB silencing (Fig. 1e, f and Supplementary Fig. 1i, j), and the 4-HNE signal colocalized with the signal of the mitochondrial marker TOM20 (translocase of outer mitochondrial membrane 20) (Fig. 1e and Supplementary Fig. 1i).

Ferroptosis induction is characterized by shrinking mitochondria with decreased crista and condensed/ruptured membranes[13,28–30]. LDHB silencing in A549 and H460 cells resulted in shrunken mitochondria with more condensed membranes, accompanied by an increase in the size of autophagosomes and necrosis-related vacuoles (Fig. 1g and Supplementary Fig. 1k), as previously observed in H460 cells after ferroptosis induction by ionizing radiation[29]. In addition, the silencing of LDHB in A549 cells was accompanied by changes in cell shape, including rounding of the cells, which resulted in individual cells looking like a single, shiny bleb (similar to one of the paired, dividing cells during mitosis) (Supplementary Fig. 1s), a phenotype previously associated with the induction of ferroptosis[31]. Since mitochondrial integrity is essential for viability[32], we determined the effect of LDHB silencing on the relative cell viability, defined as the percentage of healthy cells in a sample compared to the corresponding untreated control[33]. Indeed, the relative viability of all the cancer cell lines described above was significantly reduced by silencing LDHB (Supplementary Fig. 1l–p), as described for some of the lung cancer cell lines in our previous study[11]. Further, the reduction in colony formation upon silencing LDHB could be partially rescued by supplementation with the small molecule ferrostatin-1 (FER1), a lipid

hydroperoxyl radical scavenger, which inhibits iron-dependent lipid peroxidation[13,34] (Fig. 1h, i and Supplementary Fig. 1t). Intriguingly, supplementation with mitochondria-targeted TEMPO (mitoTEMPO), which accumulates in active mitochondria where the reduction of its active nitroxide to hydroxylamine is mainly mediated by ubiquinol[35],

also significantly rescued survival upon LDHB silencing (Fig. 1h, i) suggesting that mitochondrial lipid peroxidation significantly contributes to the reduction in colony formation upon LDHB silencing.

The accumulation of lipid peroxidation can be triggered by an excess of reactive oxygen species (ROS)[36]. However, LDHB silencing

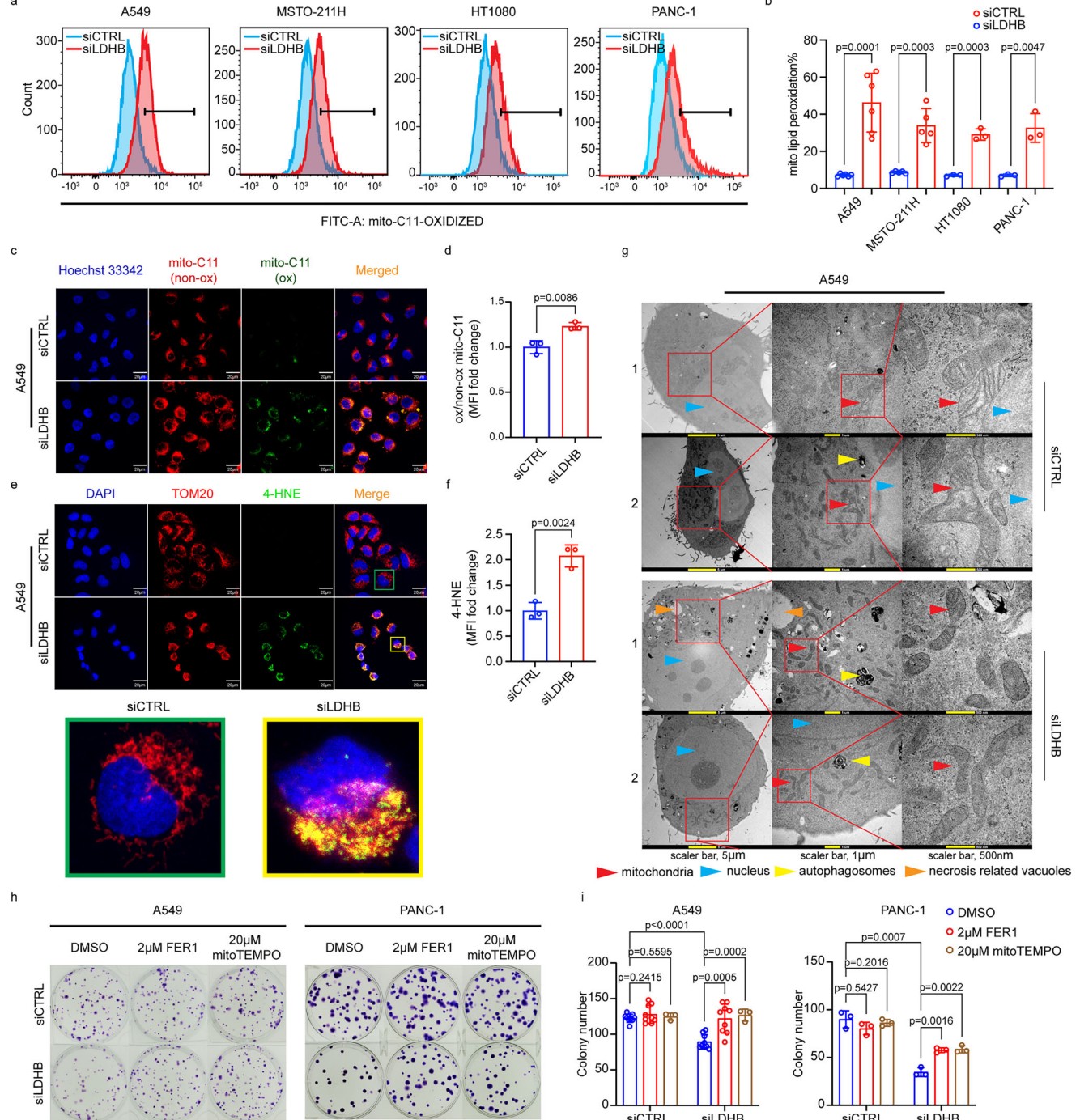

**Fig. 1 | Silencing of LDHB induces mitochondrial lipid peroxidation in cancer cells.** Measurement of mitochondrial lipid peroxidation in A549, MSTO-211H, HT1080 and PANC-1 cells after 72 h of transfection with siRNA (siCTRL) or with LDHB-targeted siRNAs (siLDHB) (**a, b**), $n = 6$, $n = 5$, $n = 3$ independent replicates for A549, MSTO-211H, HT1080, and PANC-1 respectively. Mitochondrial lipid peroxidation (**c, d** scale bar, 20 μm), 4-HNE, and TOM20 colocalization (**e, f** scale bar, 20 μm) were examined by immunofluorescence in A549 siRNAs cells after 72 h of transfection, $n = 3$ independent replicates. Transmission electron microscopy images of A549 siRNAs cells after 72 h of transfection (**g**). Images of the clonogenic

assays of A549, PANC-1 siRNAs cells at day 10 after treatment with DMSO or 2 μM ferrostatin-1 (FER1) or 20 μM mitoTEMPO (**h**). Analysis of colony numbers of A549, PANC-1 siRNAs cells at day 10 after treatment with DMSO or 2 μM ferrostatin-1 (FER1) or 20 μM mitoTEMPO, $n = 9$ independent replicates for DMSO and FER1 treated groups, $n = 3$ independent replicates for mitoTEMPO treated groups for A549 cell line. $n = 3$ independent replicates for PANC-1 cell line (**i**). Data were presented as mean ± SD. Unpaired, two-tailed $t$-test. ns no significant difference, *$P < 0.05$, **$P < 0.01$, ***$P < 0.001$, and ****$P < 0.0001$. Source data are provided as a Source Data file.

did not result in a dramatic increase in cellular ROS levels. In detail, 72 h after LDHB silencing, cellular ROS levels were not altered in A549 and HT1080 cells. Compared to ROS induction by treatment with tert-butyl hydroperoxide (TBHP), an exogenous inducer of oxidative stress[37], ROS levels were only slightly increased in PANC-1 cells and actually decreased MSTO-211H cells (Supplementary Fig. 1q, r). This is consistent with our recently published findings, which showed that total and mitochondrial ROS levels in A549 cells were not significantly increased 48 h after LDHB silencing[38].

In conclusion, LDHB silencing results in decreased viability and colony formation capacity in cancer cell lines from different genetic backgrounds and tissues of origin and is associated with increased lipid peroxidation, particularly mitochondrial lipid peroxidation, accompanied by changes in mitochondrial morphology, all characteristics associated with ferroptosis.

## LDHB suppresses mitochondria-associated ferroptosis in cancer cells

To further characterize the function of LDHB in suppressing lipid peroxidation-associated cell death, we combined LDHB silencing with RSL3 treatment, which irreversibly blocks GPX4, an essential regulator of ferroptotic cancer cell death[17]. The addition of RSL3 treatment to control transfection decreased cell viability in all 16 tested cancer cell lines (Fig. 2a, first column), which is consistent with previous data[17,20,39,40]. The normalized viability of all 16 cell lines was significantly further reduced when RSL3 treatment was combined with LDHB silencing (Fig. 2a, second column, Fig. 2c, and Supplementary Fig. 2a–d). In agreement, the alternative GPX4 inhibitor ML162 also synergistically reduced survival when combined with LDHB silencing in four tested cell lines (Supplementary Fig. 2e). To test whether the observed effects were specific to LDHB inhibition and not due to reduced overall LDH activity, we also silenced LDHA expression (Supplementary Fig. 2g). LDHA silencing did not change LDHB protein levels in the four tested cell lines (Supplementary Fig. 2g). Interestingly, LDHA silencing did not further promote the reduction in cell viability induced by RLS3, but actually led to resistance to RSL3 treatment in four cancer cell lines of different origins compared to transfected control cells (Supplementary Fig. 2f). Thus, our experiments revealed that silencing LDHB, but not LDHA, augments RSL3-induced loss of viability in cancer cells.

It was shown before that supplementation with Ferrostain-1 (FER1), which suppresses the accumulation of lipid hydroperoxides in a catalytic fashion[41], inhibited cell death induced by RSL3 treatment[13]. Interestingly, FER1 supplementation almost completely rescued the RSL3-induced reduction in cell viability and colony formation in control and siLDHB-transfected cell lines (Fig. 2a, b and Supplementary Fig. 2a–d), suggesting that the combination of RSL3 treatment and LDHB silencing results in increased levels of lipid hydroperoxides that reduce colony formation and survival.

Next, we used organelle-specific small molecule compounds to further dissect the anti-ferroptotic function of LDHB in mitochondria. The RSL3-induced reduction in cell viability based on the PrestoBlue or APH assay could be fully rescued by the supplementation with FER1 (DMSO-soluble) or the broad radical scavenger TEMPO (water-soluble) (Fig. 2c and Supplementary Fig. 2i, respectively). Interestingly, supplementation with mitochondria-targeted TEMPO (mitoTEMPO), which accumulates in active mitochondria where the reduction of its active nitroxide to hydroxylamine is mainly mediated by ubiquinol[35], also significantly rescued survival upon combined LDHB silencing and RSL3-treatment (Fig. 2c and Supplementary Fig. 2i, j) suggesting that not all, but a significant fraction of the induced lipid peroxidation occurs in the mitochondria.

At the molecular level, staining with the mitochondria-targeted lipid peroxidation probe, ox-mitoC11, revealed that RSL3 treatment at the concentration tested did not lead to a significant increase in mitochondrial lipid peroxidation in the four cell lines tested (Fig. 2d, e, column 1 versus 2), which was published for HT1080 cells before[19]. However, LDHB silencing resulted in dramatically increased mitochondrial lipid peroxidation levels compared to control cells (Fig. 2d, e, column 1 versus 6). When compared to either treatment alone, mitochondrial lipid peroxidation levels were further increased when LDHB silencing was combined with RSL3 treatment (Fig. 2d, e, column 2/6 versus 7). Therefore, we conclude that, in the absence of LDHB, RSL3 treatment indeed leads to increased mitochondrial lipid peroxidation levels, which agrees with a function of GPX4 in the mitochondria, as reviewed before[42]. The increased mitochondrial lipid peroxidation levels upon combined LDHB silencing and RSL3 treatment were suppressed by the supplementation with FER1, TEMPO, and also mitoTEMPO (Fig. 2d, e, column 7 versus 8/9/10). Previously, it has been shown that knockout of GPD2 in HeLa cells leads to increased cell death, which was associated with increased lipid peroxidation levels and was further augmented by additional RSL3 treatment[20]. Indeed, both LDHB silencing and RSL3 treatment resulted in increased cell death, which was significantly augmented by the combination thereof (Fig. 2f and Supplementary Fig. 2k). Intriguingly, the increased cell death levels upon combined LDHB silencing and RSL3 treatment were suppressed by the supplementation with FER1, TEMPO, and also mitoTEMPO (Fig. 2f and Supplementary Fig. 2k). In summary, silencing LDHB dramatically increases mitochondrial lipid peroxidation levels and cell death, which can be even further increased by additional RSL3 treatment. Thus, our results confirm our hypothesis that LDHB suppresses ferroptosis, particularly mitochondria-associated ferroptosis, in cancer cells.

## LDHB acts in parallel with GPX4 to suppress mitochondria-associated ferroptosis

We described before that LDHB is required for plasma membrane-localized SLC7A11-mediated glutathione metabolism in the tested KRAS-mutant lung cancer cells[24]. However, the SLC7A11-related function of LDHB cannot fully account for the mitochondrial phenotype observed after LDHB silencing (Fig. 1). Also, silencing LDHB did not decrease SLC7A11 expression in KRAS wild-type cancer cells; in fact, an increase in SLC7A11 was observed in HT1080 cells after LDHB silencing (Fig. 3a), but LDHB silencing still sensitized KRAS wild-type cancer cells to GPX4 inhibition (Fig. 2a). Thus, we next examined the genetic interactions between LDHB and genes known to confer anti-ferroptotic activity in the mitochondria. GPX4 protects both cytoplasmic and mitochondrial membranes from ferroptosis[42]. Interestingly, silencing of LDHB increased total cellular protein levels of GPX4 (Fig. 3a and Supplementary Fig. 3a, b). Further, after silencing LDHB, increased GPX4 expression was found to colocalize with the mitochondrial marker Cytochrome C Oxidase Subunit IV (COX IV) (Fig. 3b and Supplementary Fig. 3c). In addition, our in silico analysis of the 1739 cancer cell lines and 10967 cancer patient samples revealed a significant negative correlation between LDHB and GPX4 expression, i.e., either one of the two genes is highly expressed but rarely both together (Supplementary Fig. 3d). Our in silico analysis of single-cell sequencing data from human breast and lung adenocarcinoma tumors as well as from human pancreatic normal tissue further revealed the existence of distinct subpopulations, which feature either high LDHB or GPX4 expression but rarely both together (Supplementary Fig. 3e–g). Thus, both the analysis of the available in silico data and our experimental data indicate that GPX4 and LDHB follow a mutually exclusive expression pattern suggesting that the LDHB- and GPX4-dependent pathways function independently.

Next, we aimed to validate the hypothesis that the LDHB- and GPX4-dependent pathways function in parallel to suppress mitochondria-associated ferroptosis. To avoid potential bias due to RSL3-induced off-target effects, we used a virus-based system to induce GPX4 deletion in A549 and HT1080 cells combined with short-term silencing of LDHB (Supplementary Fig. 3h). Consistent with the

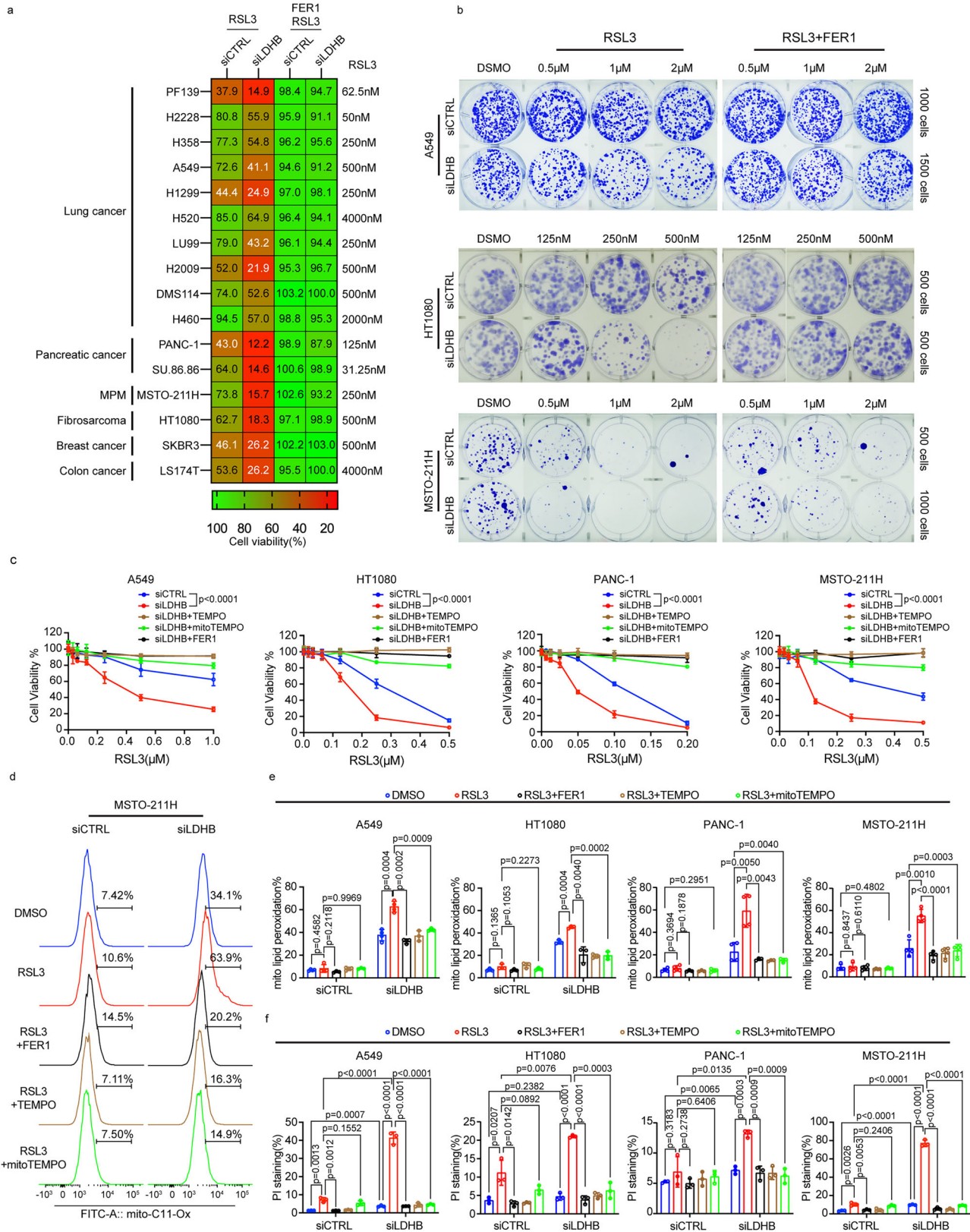

results after RSL3 treatment (Fig. 2), the combined suppression of LDHB and GPX4 expression significantly increased mitochondrial lipid peroxidation compared to either treatment alone both in HT1080 and A549 cells (Fig. 3d and Supplementary Fig. 3i, respectively). Cell death was increased upon the combined suppression of LDHB expression and GPX4 deletion in A549 cells and could be rescued by both FER1 and

mitoTEMPO treatment (Supplementary Fig. 3i, right panel). In agreement, deletion of GPX4 in HT1080 cells was accompanied by a reduction in survival and the appearance of ferroptosis-associated rounding of a fraction of the cells[31] (see also Supplementary Fig. 1s), which was exacerbated by additional LDHB silencing (Fig. 3c). Both the increase in lipid peroxidation and the reduction in survival could be

**Fig. 2 | LDHB suppresses mitochondria-associated ferroptosis in cancer cells.**
Heatmap of cell viability and images of the clonogenic assay of siRNAs cells treated with RSL3 alone or in combination with 5 μM FER1 for 48 h after pretreatment with vehicle or 5 μM FER1 for 24 h (**a**, **b**). Assessment of cell viability with PrestoBlue in A549, HT1080, PANC-1, MSTO-211H siRNAs cells treated with DMSO or RSL3 alone or in combination with 20 μM mitoTEMPO, 20 μM TEMPO, 5 μM FER1 for 48 h after pretreatment with vehicle, 20 μM mitoTEMPO, 20 μM TEMPO, 5 μM FER1 for 24 h, $n = 3$ independent replicates (**c**). Assessment of mitochondrial lipid peroxidation by flow cytometry in A549, HT1080, PANC-1, MSTO-211H siRNAs cells treated with DMSO or RSL3 alone (1 μM for A549, 0.75 μM for HT1080, and MSTO-211H cells, 0.5 μM PANC-1) or in combination with 20 μM mitoTEMPO, 20 μM TEMPO, 5 μM FER1 for 1 h after pretreatment with vehicle, 20 μM mitoTEMPO, 20 μM TEMPO, 5 μM FER1 for 24 h, $n = 4$ independent replicates for A549, PANC-1, MSTO-211H cell lines treated with DMSO and $n = 3$ independent replicates for the rest groups, $n = 3$ independent replicates for HT1080 cell line (**d**, **e**). Analysis of PI staining by flow cytometer in A549, HT1080, PANC-1, MSTO-211H cells after 72 h of transfection with siRNAs treated with DMSO or RSL3 alone (1 μM for A549, 500 nM for HT1080, 250 nM for MSTO-211H, 50 nM for PANC-1) or in combination with 5 μM FER1, 20 μM mitoTEMPO, 20 μM TEMPO for 24 h. $n = 3$ independent replicates (**f**). Data were presented as mean ± SD. Two-way ANOVA (**c**), Unpaired, two-tailed *t*-test (**e**, **f**). ns no significant difference, *$P < 0.05$, **$P < 0.01$, ***$P < 0.001$, and ****$P < 0.0001$. Source data are provided as a Source Data file.

rescued by treatment with FER1 and also mitoTEMPO after combined suppression of LDHB and GPX4 expression in HT1080 cells (Fig. 3c–e).

As an alternative approach to evaluate the interaction between LDHB and GPX4, we generated RSL3-resistant cell cultures by treating parental HT1080 with 4 μM RSL3 for ten cycles (Fig. 3f). Both LDHB and GPX4 protein expression levels were increased in RSL3-resistant HT1080 cultures, while LDHA expression remained unchanged (Fig. 3g). Indeed, LDHB silencing sensitized the RLS3 resistant cells to RSL3 treatment (Fig. 3h). Finally, overexpression of LDHB in HT1080 and COR-L105 cells resulted in dramatically reduced GPX4 expression levels, which correlated with increased resistance to RSL3 treatment (Fig. 3i, j and Supplementary Fig. 3j, k, respectively), corroborating the hypothesis that LDHB and GPX4 work in parallel to suppress mitochondria-associated ferroptosis.

GPX4 activity is dependent on glutathione, which serves as a co-factor and reducing substrate[17]. We previously showed that LDHB silencing results in reduced levels of glutathione[11], which is not effectively imported into cells[43]. However, the monoethyl ester of glutathione (GSH-mee), in which the carboxyl group of the glycine residue is esterified, is readily imported into cells and is hydrolyzed intracellularly, resulting in increased cellular levels of glutathione[43]. Indeed, supplementation with GSH-mee rescued the RSL3-induced reduction in colony formation of control HT1080 cells (Fig. 3k, top row), suggesting that GSH-mee can rescue the inhibition of GPX4. However, the reduction in colony formation upon combined RSL3-treatment and LDHB silencing was only slightly rescued by GSH-mee supplementation (Fig. 3k, bottom row), indicating that the function of LDHB required for colony formation works in parallel with GPX4 and GSH.

Finally, our in vivo experiments revealed that the combined silencing of LDHB and RSL3 treatment significantly reduced the growth and weight of A549 xenograft tumors compared with single treatments (Supplementary Fig. 3l, m). Intriguingly, at the end of the in vivo experiment, lipid peroxidation levels, as determined by 4-HNE staining, were still significantly increased after combined LDHB silencing and RSL3 treatment (Supplementary Fig. 3n, o). To exclude RSL3-mediated off-target activity, combined silencing of LDHB and GPX4 deletion also significantly reduced the growth and weight of A549 and HT1080 xenograft tumors compared to single treatments (Fig. 3l, m and Fig. 3o, p, respectively). Notably, lipid peroxidation levels remained elevated following the combined silencing of LDHB expression and GPX4 deletion. However, these levels could be restored through the administration of liproxstatin-1 (Lip1), a free radical scavenging antioxidant and thus a potent inhibitor of ferroptosis[41] (Fig. 3n, q and Supplementary Fig. 3p–s). Together, our in vitro, in silico, and in vivo data indicate that LDHB and GPX4 work in parallel to suppress lipid peroxidation, particularly in mitochondria, thereby corroborating our hypothesis that LDHB is an important contributor to suppress mitochondria-associated ferroptosis in cancer cells.

## LDHB acts in parallel with DHODH to suppress mitochondria-associated ferroptosis

Next, we studied the genetic interaction between LDHB and additional anti-ferroptotic pathways, in particular those related to mitochondria-associated ferroptosis. It was shown before that FSP1 and DHODH operate in separate cellular compartments to defend against ferroptosis, e.g., on the plasma membrane and in the mitochondria, respectively[42]. Indeed, LDHB gene expression negatively correlated with the expression of AIFM2, which encodes FSP1, in the Cancer Cell Line Encyclopedia and also in the 10967 samples included in the TCGA PanCancer Atlas (Supplementary Fig. 4k, l). Further, the reduction in viability after LDHB silencing was not significantly augmented by additional FSP1 knockout (sgFSP1) (Supplementary Fig. 4g–i). However, additional silencing of LDHB further reduced the viability of A549 and HT1080 FSP1 knockout cells after treatment with RSL3 (Fig. 4d), suggesting that the functions of LDHB and FSP1 in protecting cells from ferroptosis do not overlap.

Subsequently, we investigated the genetic interaction of LDHB and dihydroorotate dehydrogenase (DHODH), which was shown to reduce CoQ in the inner mitochondrial membrane and can compensate for the loss of mitochondrial GPX4 to detoxify mitochondrial lipid peroxidation[19]. A549 and HT1080 cells harboring a genetic knockout of DHODH (sgDHODH) (Supplementary Fig. 4e, f) were cultured in uridine-supplemented media to prevent them from undergoing ferroptosis-independent cell death, as previously published[19]. DHODH deletion did not significantly affect LDHB expression (Supplementary Fig. 4e, f). Compared to both single alterations, the combination of DHODH deletion and LDHB silencing significantly increased mitochondrial lipid peroxidation levels (Fig. 4a, b and Supplementary Fig. 4a, b), which inversely correlated with cell viability and survival of A549 and HT1080 cells (Fig. 4b and Supplementary Fig. 4c, d). After RSL3 treatment, the viability of A549 and HT1080 cells was significantly reduced by the combination of DHODH deletion and LDHB silencing compared to both single alterations and could be completely rescued by FER1 supplementation (Fig. 4c). Thus, we concluded that GPX4, DHODH, and LDHB work in parallel to suppress mitochondrial lipid peroxidation and thus protect against mitochondria-associated ferroptosis.

## Silencing of LDHB is associated with the anti-ferroptotic function of ubiquinol

To investigate in more detail the molecular mechanisms underlying the role of LDHB in suppressing mitochondria-associated ferroptosis, we treated cells with ferroptosis inducer 56 (FIN56), which not only depletes GPX4 but also activates squalene synthase, thereby depleting CoQ synthesis[44]. Interestingly, silencing of LDHB did not further sensitize PANC-1 and MSTO-211H cancer cells to FIN56 treatment compared with control silencing (Fig. 4e and Supplementary Fig. 4m, respectively), revealing that the anti-ferroptotic function of LDHB overlaps with the functions that are inhibited by FIN56, e.g., the GPX4- and CoQ-dependent pathways[44]. Since the reduction in viability upon combined LDHB silencing and FIN56 treatment could be rescued by FER1 treatment, the overlapping function has to be related to lipid peroxidation (Fig. 4e and Supplementary Fig. 4m, respectively). Interestingly, compared to control cells, LDHB silencing further sensitized A549 cells to FIN56 treatment (Supplementary Fig. 4n). The decreased cell viability induced by LDHB silencing in combination with

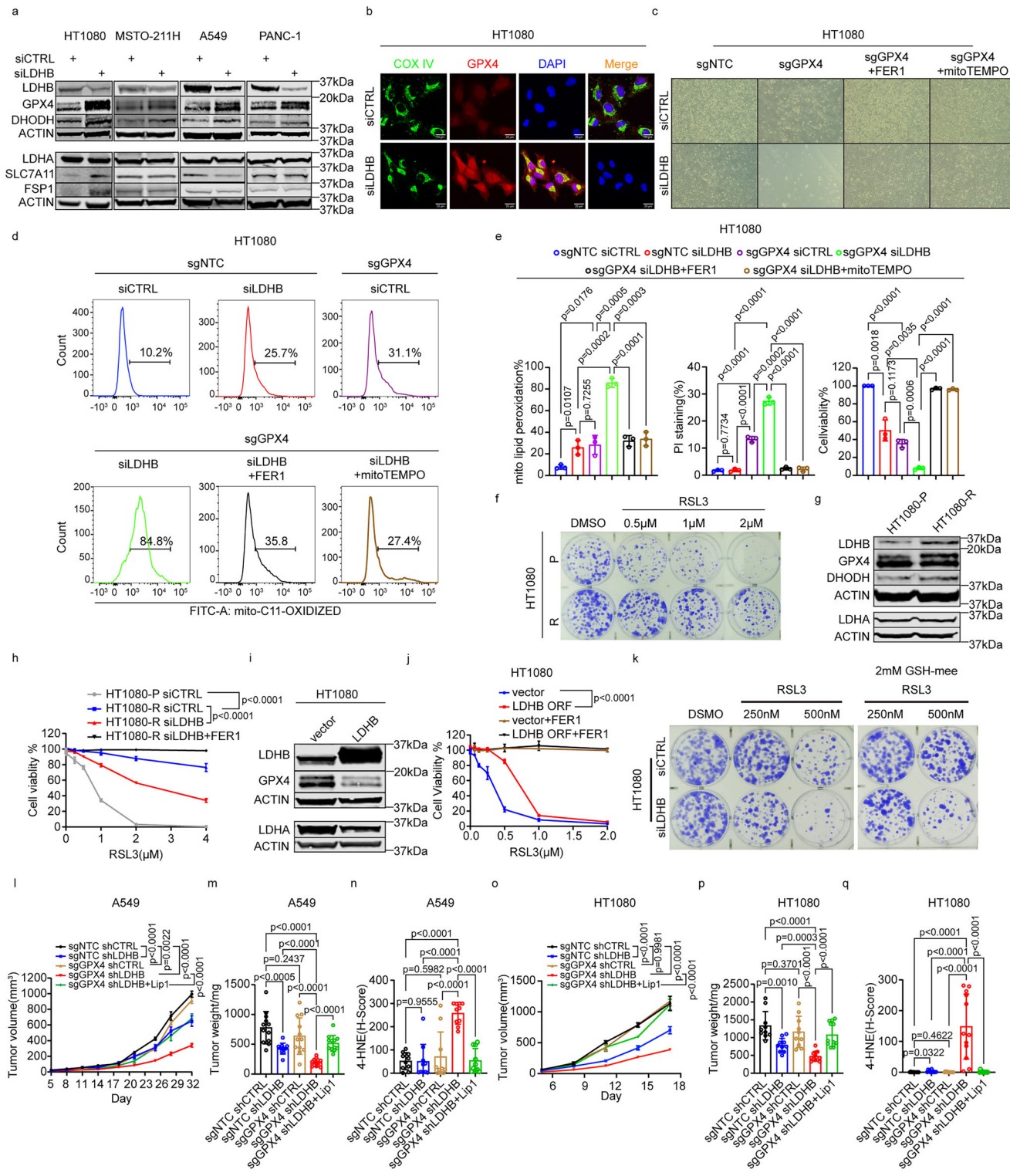

FIN56 treatment can only be partially rescued by FER1 (Supplementary Fig. 4n), suggesting that LDHB silencing in combination with FIN56 treatment may induce not only ferroptosis but also other types of cellular stress in A549 cells.

4-chlorobenzoic acid (4CBA) blocks the tyrosine- and mevalonate-dependent synthesis of CoQ[45]. Consistent with the results after FIN56 treatment (Fig. 4e), cell viability after combined treatment with 4CBA and RSL3 i.e., combined blockade of GPX4 and CoQ synthesis, was not further reduced by additional LDHB silencing in PANC-1 cells (Fig. 4f), suggesting that the effect of LDHB silencing is at least partially

redundant with 4CBA treatment. Also, additional LDHB silencing in *KRAS*-mutated A549 cells further reduced cell viability after combined treatment with 4CBA and RSL3 (Supplementary Fig. 4o). Thus, the increased sensitivity to RSL3 induced by LDHB silencing in A549 cells is not solely dependent on the CoQ-associated defense mechanism, which is consistent with our previously published study showing that LDHB silencing also affects the GSH-associated defense mechanism in *KRAS*-mutated lung cancer[24].

Concerning the anti-ferroptotic function of CoQ in mitochondria, it was shown before that DHODH reduces ubiquinone to ubiquinol in

**Fig. 3 | LDHB acts in parallel with GPX4 to suppress mitochondria-associated ferroptosis.** Immunoblot analysis of LDHB, GPX4, DHODH, LDHA, SLC7A11, and FSP1 in HT1080, MSTO-211H, A549, PANC-1 siRNAs cells, $n = 3–6$ independent repeats, and the statistical analysis was shown in Supplementary Fig. 3a, b (**a**). Colocalization analysis by immunofluorescence in HT1080 siRNAs cells after 72 h of transfection (**b**, scale bar, 20 μm). Images of microscopic analysis (**c**, 20x objective) and assessment of mitochondrial lipid peroxidation by flow cytometry in HT1080 sgNTC and sgGPX4 cells after 72 h of transfection with siRNAs, $n = 3$ independent replicates (**d**, **e**). Clonogenic assays of parental HT1080 (HT1080-P) and HT1080 cells treated with 4 μM RSL3 after ten cycles (HT1080-R) (**f**). Immunoblot analysis of LDHB, GPX4, DHODH, and LDHA in HT1080 parental and RSL3 resistant cells. The experiment was repeated three times, yielding consistent results (**g**). Cell viability of HT1080-P and HT1080-R cells after transfection with siRNAs treated with DMSO or RSL3 alone or in combination with 5 μM FER1 for 48 h, following pretreatment with vehicle, 5 μM FER1 for 24 h, $n = 3$ independent replicates (**h**). Immunoblot analysis of LDHB, GPX4, LDHA in HT1080 control and LDHB overexpression cells (**i**). Cell viability assays of HT1080 cells and HT1080 LDHB ORF cells treated with RSL3 with or without 5 μM FER1 for 48 h, following pretreatment with vehicle, 5 μM FER1 for 24 h. $n = 3$ independent replicates (**j**). Clonogenic assay of HT1080 siRNAs cells treated with DMSO or RSL3 alone or in combination with 2 mM glutathione reduced ethyl ester (GSH-mee) for 48 h after pretreatment with vehicle, 2 mM GSH-mee for 24 h (**k**). Tumor volume and weight of A549 sgNTC shCTRL ($n = 12$), sgNTC shLDHB ($n = 11$), sgGPX4 shCTRL ($n = 12$), sgGPX4 shLDHB ($n = 12$), sgGPX4 shLDHB treated with liproxstatin-1 ($n = 12$) xenograft tumors from different mice (**l**, **m**), tumor volume and weight of HT1080 sgNTC shCTRL ($n = 10$), sgNTC shLDHB ($n = 10$), sgGPX4 shCTRL ($n = 10$), sgGPX4 shLDHB ($n = 12$), sgGPX4 shLDHB treated with liproxstatin-1 ($n = 12$) xenograft tumors from different mice (**o**, **p**), 4-HNE expression of A549 and HT1080 sgNTC shCTRL, sgNTC shLDHB, sgGPX4 shCTRL, and sgGPX4 shLDHB xenograft tumors treated with or without liproxstatin-1, $n = 10$ or $n = 12$ different regions from different tumors from different mice respectively (**n**, **q**). Data were presented as mean ± SEM (**l**, **o**) or mean ± SD (**e**, **h**, **j**, **m**, **n**, **p**, **q**). Two-way ANOVA (**h**, **j**, **l**, **o**), Unpaired, two-tailed $t$-test (**e**, **m**, **n**, **p**, **q**). ns no significant difference, *$P < 0.05$, **$P < 0.01$, ***$P < 0.001$, and ****$P < 0.0001$. Source data are provided as a Source Data file.

the inner mitochondrial membrane, thereby efficiently inhibiting mitochondrial lipid peroxidation[19]. Thus, in analogy to the anti-ferroptotic function of DHODH[19], we proposed the following working model: LDHB reduces ubiquinone to ubiquinol in the inner mitochondrial membrane, thereby suppressing mitochondria-associated ferroptosis (Fig. 4j). Indeed, LDHB protein levels and activity were increased in mitochondrial lysates compared with total cell lysates (Supplementary Fig. 4w, x), which agrees with a previous study[25]. In addition, LDHB protein is present in the fraction containing the mitochondrial matrix and inner membrane but not in the fraction containing the outer mitochondrial membrane marker TOM20 (Supplementary Fig. 4y). Also in agreement with our working model, silencing of LDHB increased the ratio of CoQ to CoQH2 (Fig. 4g and Supplementary Fig. 4p), as shown before, after DHODH inhibition and GPD2 knockout[19,20].

To further substantiate our working model, we re-analyzed the data obtained in our previous study showing that the silencing of LDHB results in a significant reduction in mitochondrial metabolism[11]. Specifically, we applied transcriptomics-driven metabolic pathway analysis (TDMPA), a method that uses genome-scale metabolic models to calculate perturbations of enzymatic reactions from gene expression data[46]. Although ubiquinol has multiple roles in cellular metabolism, it is most known for its ability to shuttle electrons between mitochondrial electron transport chain complexes, driving ATP synthesis via oxidative phosphorylation[47]. Our working model suggests that LDHB silencing should interfere with the direct interaction of LDHB and ubiquinol at the inner mitochondrial membrane, where oxidative phosphorylation also occurs. Intriguingly, TDMPA revealed that "oxidative phosphorylation" is the only significantly dysregulated metabolic pathway in A549 cells and the most significantly dysregulated in H358 cells after LDHB silencing (Supplementary Fig. 4u).

It has previously been shown that FSP1 can act as a glutathione-independent CoQ oxidoreductase to inhibit ferroptosis[40]. In this study, the authors performed an in vitro NADH consumption assay in the presence of different electron acceptor molecules and showed that purified human FSP1 consumes NADH in the presence of CoQ1 and CoQ10. Consequently, we performed the same NADH consumption assay but substituted human FSP1 with commercially available human LDHB. In detail, in the presence of LDHB, NADH consumption was increased by CoQ in a dose-dependent manner (Fig. 4h). In control reactions with a fixed concentration of CoQ, increasing the concentration of LDHB also increased NADH consumption (Supplementary Fig. 4r). Furthermore, the addition of sodium lactate increased the rate of NADH consumption over time in the presence of 200 and 800 μM CoQ (Supplementary Fig. 4q). Thus, our experiments demonstrated that LDHB catalyzes the transfer of reducing

equivalents from NADH to CoQ and that the efficiency of this reaction can be increased by the addition of lactate. Relevant in this context, putative LDH(A) was identified in a screen as a protein that binds pyrroloquinoline quinone (PQQ), a redox-active o-quinone that is an important nutrient for various biochemical processes in mammals[48]. Our reanalysis of the available data revealed that of the four peptides attributed to LDH(A) by the authors, the sequence of peptide #4 (VIGSGCNLDSAR) also aligned to 100% with LDHB (https://www.uniprot.org/uniprotkb/P07195/entry#sequences, aligned via https://blast.ncbi.nlm.nih.gov/Blast.cgi). The authors showed that PQQ-bound LDH(A), in the presence of NAD+, catalyzes the conversion of lactate to pyruvate, thereby generating NADH. In the presence of NADH, PQQ-bound LDH(A) catalyzes the oxidation of the bound PQQ to PQQH2, generating NAD+. We found that NADH was efficiently generated when lactate was combined with LDHB and NAD+ (Fig. 4i). Furthermore, when CoQ was added to this reaction, NADH levels were reduced in a CoQ-dose-dependent manner (Fig. 4i), suggesting that the NADH generated by the LDHB-mediated oxidation of lactate is subsequently consumed in a second reaction, e.g., the NADH-dependent reduction of CoQ to CoQH2, which is also mediated by LDHB (Fig. 4j). In summary, our in vitro results support our hypothesis that LDHB, analogous to FSP1, transports reducing equivalents of NADH to CoQ into the lipid bilayer, thereby facilitating CoQH2-mediated inhibition of lipid peroxidation.

It was shown before that treatment with a mitochondria-targeted analog of CoQH2 (mitoCoQH2), protected DHODH knockout cells against mitochondrial lipid peroxidation and ferroptosis induced by GPX4 inhibitors, i.e., RSL3 and ML162[19]. Indeed, supplementation with mitoCoQH2 dramatically suppressed mitochondrial lipid peroxidation and cell death after combined RSL3 treatment and LDHB silencing in the four cell lines tested (Fig. 4k–m and Supplementary Fig. 4v). In agreement, supplementation with mitoCoQH2 fully rescued viability after combined RSL3 treatment and LDHB silencing whereas supplementation with mitoCoQ and CoQ10 was less effective (Fig. 4m and Supplementary Fig. 4s). Thus, our experiments supported our hypothesis that LDHB suppresses mitochondria-associated ferroptosis in a ubiquinol-dependent manner.

### Targeting LDHB sensitizes tumor cells to radiotherapy by enhancing mitochondria-associated ferroptosis

It has been shown previously that exposure of cancer cells to ionizing radiation resulted in the induction of ferroptosis[29], which was associated with morphological changes similar to those observed in this study after silencing LDHB (Fig. 1). Indeed, in all four cancer cell lines tested, RT resulted in extensive mitochondrial lipid peroxidation, which was further enhanced when combined with LDHB silencing

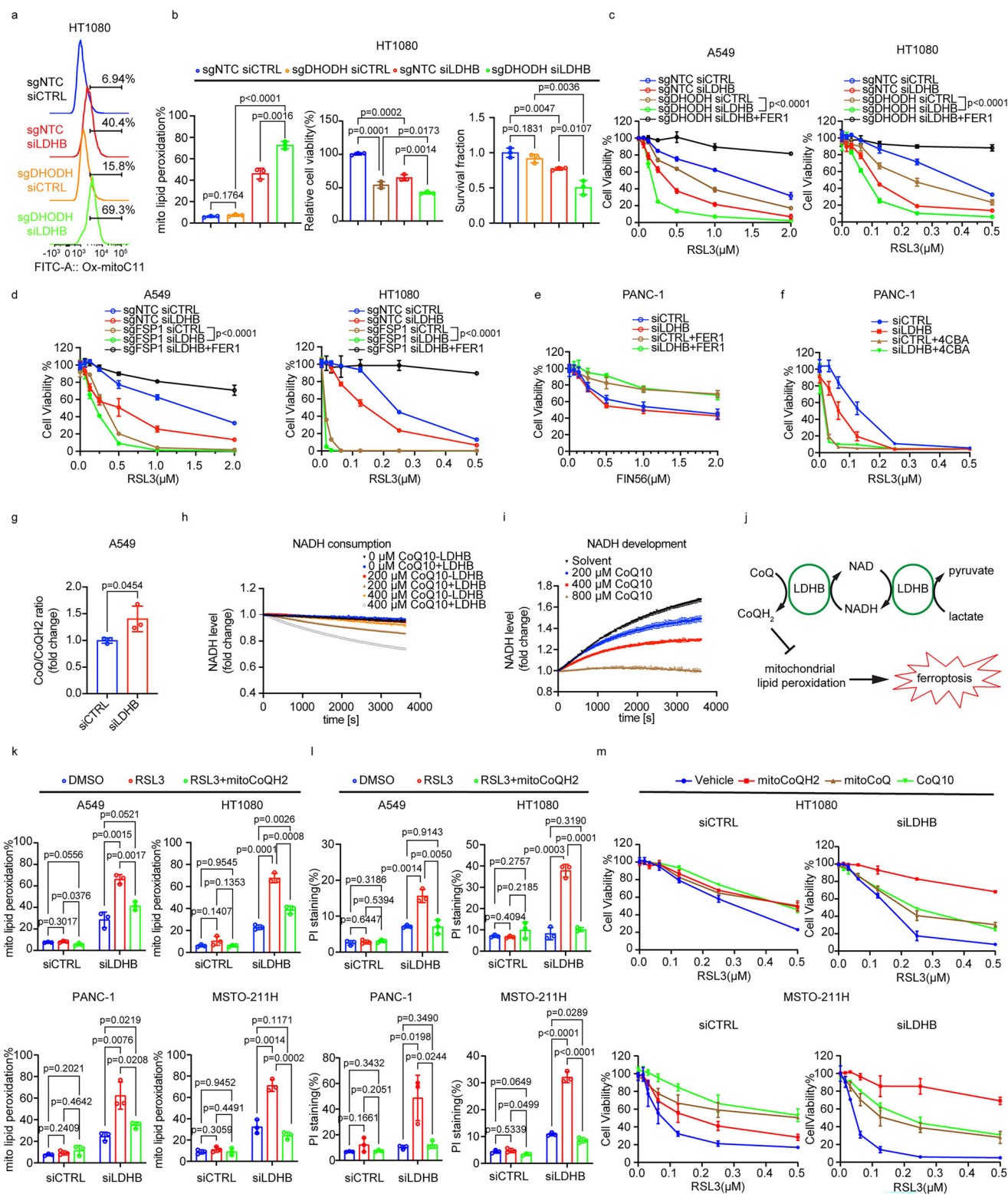

(Fig. 5a–c and Supplementary Fig. 5a). Further, combined RT and short-term LDHB silencing consistently reduced fractional survival in all twelve tested cancer cell lines from different genetic backgrounds and tissues of origin (Fig. 5d and Supplementary Fig. 5b–m). Long-term LDHB silencing also rendered A549 cells sensitive to RT, reducing colony formation (also H838 cells) and cell growth over time (Fig. 5e, f and Supplementary Fig. 5n–q). Finally, the survival fraction after RT, either alone or in combination with LDHB silencing, was rescued by the supplementation with mitoTEMPO revealing that not only LDHB

silencing but also RT induces mitochondrial lipid peroxidation (Fig. 5e, f). In conclusion, our in vitro experiments showed that RT results in extensive mitochondrial lipid peroxidation, which LDHB silencing can further enhance, revealing that LDHB protects cancer cells from RT-induced mitochondria-associated ferroptosis.

A previous analysis of a cohort of 540 lung cancer patients from the TCGA database showed that high LDHB expression was associated with shorter survival[49]. In addition, high LDHB expression was significantly associated with shorter overall survival (Fig. 5k) and median

**Fig. 4 | LDHB suppresses ferroptosis by regulating the reduction of CoQH2 in mitochondria.** Analysis of mitochondrial lipid peroxidation in HT1080 sgNTC, sgDHODH cells 72 h after transfection with siRNAs, $n = 3$ independent replicates (**a, b**). Cell viability of A549 and HT1080 sgNTC, sgDHODH or sgFSP1 cells after transfection with siRNAs treated with DMSO or RSL3 alone or in combination with 5 μM FER1 for 48 h, following pretreatment with vehicle, 5 μM FER1 for 24 h, $n = 3$ or $n = 4$ independent replicates for A549 sgDHODH cells and HT1080sgDHODH, A549sgFSP1, and HT1080sgFSP1 cells, respectively (**c, d**). Cell viability of PANC-1 transfected cells treated with DMSO or FIN56 alone or in combination with 5 μM FER1 for 48 h, following pretreatment with vehicle, 5 μM FER1 for 24 h, or treated with DMSO or RSL3 alone or in combination with 5 mM 4-carboxybenzaldehyde (4CBA) for 48 h, following pretreatment with vehicle, 5 mM 4CBA for 24 h, $n = 3$ (**e**) or 4 independent replicates (**f**). CoQ and CoQH2 analysis of A549 cells 72 h after transfection with siRNAs, $n = 3$ independent samples (**g**). NADH consumption assay (A340 nm) in TBS buffer containing different concentrations of CoQ10 with or without recombinant human LDHB. Representative curves from an independent test with three technical replicates are shown. Tests were performed independently three times (**h**). NADH development assay (A340 nm) in glycine buffer containing 1 mM lactate, 216 mM hydrazine, and 500 μM NAD with different concentrations of CoQ10. Representative curves from an independent test with three technical replicates are shown. Tests were performed independently three times (**i**). Diagram illustrating how LDHB inhibits mitochondrial lipid peroxidation (**j**). Assessment of mitochondrial lipid peroxidation (**k**) and PI staining (**l**) by flow cytometry in A549, HT1080, PANC-1, and MSTO-211H cells after 72 h of transfection with siRNAs treated with DMSO or RSL3 alone (1 μM for A549, 0.75 μM for HT1080 and MSTO-211H cells, 0.5 μM PANC-1) or in combination with 500 nM mitoCoQH2 for 1 h (**k**) or 5–6 h (**l**), $n = 3$ independent replicates. Cell viability of HT1080 and MSTO-211H transfected cells treated with DMSO or RSL3 alone or in combination with 100 nM mitoCoQH2 or 100 nM mitoCoQ or 10 μM CoQ10, for 48 h, following pretreatment with vehicle, 100 nM mitoCoQH2 or 100 nM mitoCoQ or 10 μM CoQ10 for 24 h, $n = 3$ independent replicates (**m**). Data were presented as mean ± SD. Two-way ANOVA (**c, d**). Unpaired, two-tailed $t$-test (**b, g, k, l**). ns no significant difference, *$P < 0.05$, **$P < 0.01$, ***$P < 0.001$, and ****$P < 0.0001$. Source data are provided as a Source Data file.

survival (Supplementary Fig. 5v) in 65 lung cancer patients who had received radiotherapy. To test whether targeting LDHB also sensitizes cancer cells to radiotherapy in vivo, we treated human lung cancer xenograft tumors and orthotopic lung tumors with local radiotherapy (Supplementary Fig. 5t, u). Indeed, the growth of NSCLC A549 xenograft tumors in immunodeficient mice was only slightly delayed by a single treatment with 10 Gy IR, further delayed by LDHB silencing, and most delayed by the combined treatment (Fig. 5g), which also resulted in significantly increased lipid peroxidation levels at the end of the experiment (Supplementary Fig. 5r, s). To study the effect of LDHB in an immunocompetent and orthotopic setting, we used our established genetically engineered mouse model for NSCLC that combines an *LDHB* deletion allele with the inducible model of lung adenocarcinoma driven by concomitant loss of *p53* (also known as Trp53) and expression of oncogenic *KRAS* (G12D) (KP)[11]. A reduction in viral titers administered for tumor induction allowed us to monitor individual tumor nodules over time using micro-computed tomography (microCT) (Fig. 5i). Indeed, LDHB knockout (Ldhb −/−) significantly reduced the growth of individual tumor nodules over time, as did treatment with 8 Gy IR daily for three consecutive days (Fig. 5h). However, only the combination of RT and LDHB deletion resulted in complete suppression of tumor growth during the experimental period (Fig. 5h). In detail, when combined with RT, shrinkage was observed in 9 out of 13 and only 1 out of 17 individual tumor nodules in the LDHB knockout versus wild-type background, respectively (Fig. 5j). Summarizing our in vivo experiments, silencing or knockout of LDHB dramatically reduces tumor growth and increases the radiosensitivity of human xenografts and an immunocompetent orthotopic lung tumor model.

In summary, our study revealed that silencing LDHB results in mitochondrial lipid peroxidation in a wide range of cancer cells, corroborating previous findings by others and our own that LDHB is mainly localized in mitochondria. Interestingly, the primary anti-ferroptotic function of LDHB is associated with the anti-ferroptotic function of CoQH2 and thus acts in parallel with the mitochondrial enzyme DHODH and also with the GPX4-dependent defense mechanism. We further showed that the anti-ferroptotic function of LDHB can be exploited to sensitize cancer cells to radiotherapy, a known inducer of ferroptosis.

## Discussion

All 16 cancer cell lines tested in this study were sensitive to LDHB silencing (Fig. 1h, i and Supplementary Fig. 1l–p, t). This agrees with a previous study, which revealed that sensitivity to LDHB silencing positively correlates with the LDHB protein expression level but is not limited to a specific genetic subset[9]. However, it will be interesting to elucidate the underlying genetic and molecular network that regulates LDHB expression and, thus, sensitivity to LDHB inhibition. Indeed, our previous study showed that LDHB silencing increased STAT3 expression[11], a critical tumorigenic driver in many cancers, which was shown to trans-activate LDHB gene expression[50].

We observed that the dramatic increase in mitochondrial lipid peroxidation induced by LDHB silencing is not only associated with shrunken mitochondria featuring decreased crista and condensed/ruptured membranes, a phenotype associated with the induction of ferroptosis[28,29] but also with the accumulation of autophagosomes (Fig. 1g). Autophagosomes are fused with lysosomes during mitophagy, which serves to degrade damaged mitochondria[51]. It was shown before that LDHB controls lysosome activity and autophagy in cancer cells[10]. Thus, it is critical to elucidate if the accumulation of autophagosomes upon LDHB silencing is due to an increased frequency of damaged mitochondria, a deficiency in lysosome-dependent mitophagy, or a combination thereof.

It was shown very recently that lactate is mainly metabolized in the mitochondria[52]. However, although there is ample experimental evidence for the combustion of lactate in the mitochondria, the exact molecular mechanism, particularly the exact localization, remains disputed, as recently discussed elsewhere[53]. Our previous analysis by immunofluorescence microscopy showed that LDHB colocalizes with mitochondria[11], which is in agreement with a previous study[25]. Indeed, our cell fractionation experiments revealed that LDHB is present in the fraction containing the inner mitochondrial membrane and the mitochondrial matrix (Supplementary Fig. 4y). Intriguingly, LDHA is predominantly present in the cytoplasmic fraction[25,52]. Thus, the compartmentalization of LDHA and LDHB is consistent with the concept of an intracellular lactate shuttle[54]. However, further studies on the precise localization and interaction of CoQH2-generating mitochondrial enzymes and LDHB are needed to determine their specific contribution to the anti-ferroptotic activity in mitochondria.

It has been shown before that the mitochondrial inner membrane enzyme DHODH suppresses ferroptosis in parallel with GPX[19]. Similarly, our experiments revealed that LDHB silencing synergistically reduces viability when combined with GPX4 inhibition by RSL3 treatment (Fig. 2). Interestingly, mitoTEMPO not only restored viability after combined LDHB silencing and RSL3 treatment (Fig. 2c), but also efficiently reduced the mitochondrial lipid peroxidation levels indicating that a large fraction of the total lipid peroxidation signal observed after LDHB silencing (Supplementary Fig. 1b–d) is derived from mitochondrial lipid peroxidation. However, LDHB silencing affects more than just lipid peroxidation, as the reduction in colony formation was only partially rescued by FER1 and mitoTEMPO treatment (Fig. 1h, i). This is consistent with our previous data showing that LDHB silencing broadly impacts mitochondrial metabolism[11]. Interestingly, not only the silencing of LDHB but also the inhibition of GPX4

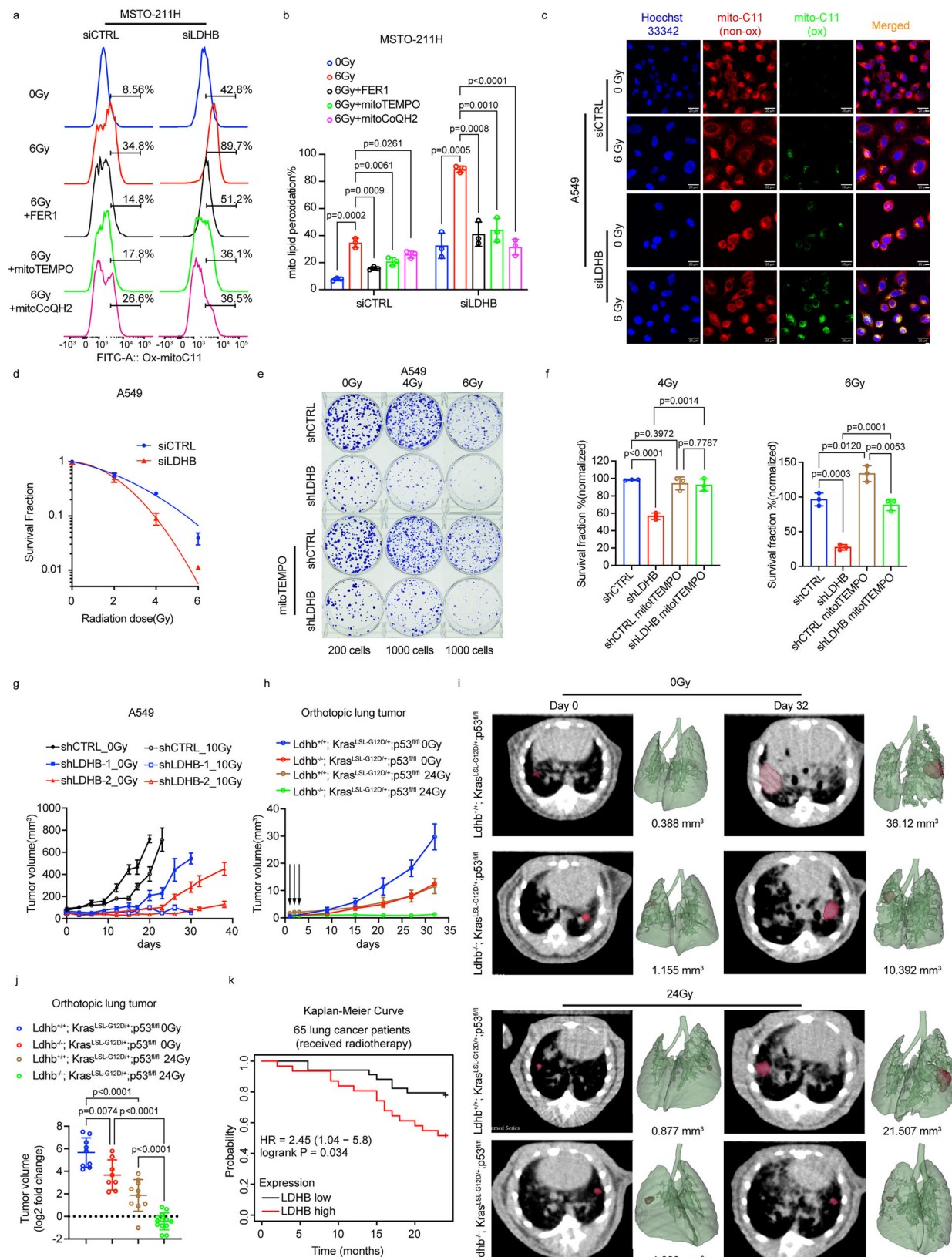

results in significant mitochondrial lipid peroxidation, which is in agreement with the known mitochondrial function of GPX4[19]. However, further experiments are needed to determine the exact relative contributions and to uncover the molecular mechanism behind the observed increase in total cellular GPX4 protein levels following LDHB silencing (Fig. 3a and Supplementary Fig. 3a, b). Intriguingly, the

expression of GPD2 (and also AIFM2, which encodes FSP1) showed a mutually exclusive expression pattern, whereas DHODH showed a positive correlation with LDHB in a large collection of cancer cell lines (Supplementary Fig. 4k, l, z). Finally, defined distinctions exist between different ferroptosis inducers[55]. Thus, it would be interesting to test how LDHB affects the response of cancer cells to ferroptosis induction

**Fig. 5 | Inhibition of LDHB sensitizes tumor cells to radiotherapy by enhancing mitochondria-associated ferroptosis.** Mitochondrial lipid peroxidation was assessed by flow cytometry in MSTO-211H after 24 h of irradiation with 6 Gy, following pretreatment with 5 μM FER1, 20 μM mitoTEMPO, 500 nM mitoCoQH2 for 5 h, $n = 3$ independent replicates (**a**, **b**). Mitochondrial lipid peroxidation was assessed by immunofluorescence in A549 siRNAs cells after 24 h of irradiation with 6 Gy, $n = 3$ random fields (**c** scale bar, 20 μm). Clonogenic survival curves for A549 siRNAs cells after irradiation with doses from 0 to 6 Gy, data were normalized to the corresponding unirradiated control group, $n = 3$ independent replicates (**d**). Representative images and analysis of clonogenic survival assay of A549 shCTRL, shLDHB cells irradiated with doses at 0, 4, 6 Gy alone or in combination with 20 μM mitoTEMPO, following pretreatment with vehicle, 20 μM mitoTEMPO for 24 h, $n = 3$ independent replicates (**e**, **f**). Volume of A549 shCTRL 0 Gy ($n = 10$), shLDHB-10 Gy ($n = 8$), shLDHB-20 Gy ($n = 12$) and 10 Gy irradiated xenograft tumors ($n = 10$) (**g**). Volume (**h**) and representative microCT images (**i**) of Ldhb[+/+]; Kras[LSL-G12D/+];p53[fl/fl] (Ldhb WT) and Ldhb[-/-]; Kras[LSL-G12D/+];p53[fl/fl] (Ldhb KO) lung tumors (red) at different time points after local irradiation with 0 Gy or 24 Gy, Ldhb WT 0 Gy ($n = 9$), Ldhb KO 0 Gy ($n = 8$), Ldhb WT 24 Gy ($n = 10$), Ldhb KO 10 Gy ($n = 12$). The log2 fold change in tumor volume after 27 days of local irradiation with 0 Gy or 24 Gy, Ldhb WT 0 Gy ($n = 9$), Ldhb KO 0 Gy ($n = 8$), Ldhb WT 24 Gy ($n = 10$), Ldhb KO 10 Gy ($n = 12$) (**j**). Kaplan–Meier overall survival analysis of high ($n = 31$) and low ($n = 34$) LDHB groups in lung cancer patients who received radiotherapy (**k**). Data were presented as mean ± SD (**b**, **d**–**f**, **j**) or as mean ± SEM (**g**, **h**). Unpaired, two-tailed $t$-test. ns no significant difference, $^*P < 0.05$, $^{**}P < 0.01$, $^{***}P < 0.001$, and $^{****}P < 0.0001$. Source data are provided as a Source Data file.

---

by cysteine-starvation, which was shown to also affect mitochondrial oxidative metabolism[21].

The question has been raised regarding which CoQH2-generating mitochondrial enzymes other than DHODH and GPD2 regulate ferroptosis[19,20]. it was reported that CoQ serves as the electron acceptor for at least seven mitochondrial inner membrane dehydrogenases next to complexes I and II of the electron transport system[18]. Our experiments showed that the consumption of the reducing agent NADH increased when LDHB was combined with increasing concentrations of CoQ, suggesting that LDHB catalyzes the transfer of reducing equivalents from NADH to CoQ (Fig. 4j). Interestingly, the consumption of the reducing agent NADH was further increased by the addition of lactate (Supplementary Fig. 4q), which is also a reducing agent. This observation can be explained by two scenarios. First, lactate binding increases the catalytic activity of LDHB, thereby enhancing the NADH-dependent reduction of CoQ. In analogy to the proposed mechanism for LDH(A) mentioned above[48], the second scenario involves three consecutive steps (Fig. 4j): In the first step, lactate oxidation drives the reduction of NAD+ bound to LDHB and CoQ, resulting in the release of pyruvate, generating NADH/PQQ bound to LDHB. In the second step, LDHB catalyzes the NADH-dependent oxidation of CoQ to CoQH2. In a third step, CoQH2 bound to LDHB and NAD+ undergoes aerobic auto-oxidation to CoQ, as previously shown for PQQH2 bound to LDH(A), in the presence of NAD+[48] completing the cycle and resulting in NAD+ and CoQ bound to LDHB. Overall, this LDHB-mediated three-step reaction would result in the transfer of reducing equivalents from lactate to CoQH2. Indeed, this LDHB-mediated three-step reaction would also explain why NADH production by LDHB-mediated lactate oxidation in the presence of NAD+ decreased in a dose-dependent manner upon addition of CoQ, i.e., the newly formed NADH (step 1) is immediately oxidized to NAD+ (step 2) in the presence of CoQ (Fig. 4j). However, we cannot formally exclude the possibility that CoQ in the presence of NAD+ blocks the oxidation of lactate and thus the production of NADH. Further experiments will be necessary to elucidate the exact details of the molecular reactions.

Although our data suggest that LDHB suppresses ferroptosis via the production of mitochondrial CoQH2, we cannot exclude the possibility of other potential mechanisms, as recently also discussed for GPD2[20]. In detail, our previous study revealed that LDHB silencing dramatically decreases total cellular NADH levels[11]. Cytosolic NADH oxidation as part of glycolytic NAD+ recycling has been recently linked to PUFA-PL desaturation[56]. Thus, silencing LDHB might increase PUFA-PL levels by decreasing NADH levels, thereby fostering ferroptosis. In addition, in a companion study submitted for publication, we describe that LDHB is required for glutathione metabolism mediated by SLC7A11, which is localized at the plasma membrane[24]. Further, LDHB has been shown to control lysosome activity[10], and a readthrough-extended version of LDHB is targeted to the peroxisome[57]. Lysosomes and peroxisomes also contribute to ferroptosis[58]. Therefore, it will be interesting to determine how and to what extent the LDHB-mediated pro- and anti-ferroptotic functions associated with the lysosomes, peroxisomes, plasma membrane/cytoplasm, and mitochondria contribute to the total cellular ferroptotic activity.

Our study agrees with previous studies, which identified LDHB as a potential therapeutic target[8–10,26,59]. Further, our study revealed that silencing LDHB augments the efficiency of RT. Indeed, it was recently shown that the combined deletion of LDHA and LDHB in human glioblastoma xenografts prolongs survival after RT[60]. Excitingly, a natural micropeptide that localizes to mitochondria where it interacts with LDHA and LDHB to prevent the conversion of lactate to pyruvate was also recently identified and shown to inhibit the growth and tumorigenicity of patient-derived primary glioblastoma cells[26]. In addition, the discovery of the first specific LDHB inhibitor was recently reported[61]. Thus, our study adds to the growing body of evidence supporting the clinical translation of targeting LDHB alone or combined with radiotherapy and other ferroptosis-inducing therapies.

## Methods

### Ethics approval and consent to participate

Mouse studies were approved by the Ethics Commission of the Canton of Bern, Switzerland (license (BE49_2022), and all experiments were conducted in accordance with the animal guidelines and protocols of the University of Bern, Switzerland.

### Cell culture

The cell culture was performed as described previously[11]. In brief, all cell lines were obtained from the American Type Culture Collection (Manassas, VA, USA), except patient-derived primary LUAD cells PF139 were established as recently reported[62]. RPMI medium (Sigma-Aldrich 8758) or DMEM/F12 (Life Technologies 21331020;) supplemented with 22 mM L-glutamine (Life Technologies 25030024); 10% fetal bovine serum/FBS (Life Technologies 10270106) and 1% penicillin/streptomycin solution (P/S, Sigma-Aldrich P0781) were used for cell culture, except for DHODH knockout cells, which were cultured with the additional supplement of 50 μM uridine at 37 °C in a humidified incubator containing 5% $CO_2$. Cell numbers were determined with an Invitrogen Countess 3 Automated Cell Counter using a hemocytometer and 0.1% trypan blue (Thermo Fisher Scientific 15250061) to exclude dead cells. All cell lines were seeded in 96-well plates for cell viability assay and in six-well plates for lipid peroxidation measurement and clonogenic survival assay. Cells treated with ferroptosis inducers including RSL3 (Chemscene CS-5650), ML162 (Chemscene CS-0017910), FIN56 (Selleck Chemicals S8254); ferroptosis inhibitors including Ferrostatin-1 (Chemscene CS-0019733); antioxidants including Glutathione ethyl ester (Cayman Chemical Company 14953), 2,2,6,6-Tetramethylpiperidine 1-oxyl (TEMPO, Sigma-Aldrich 214001-1 G), mitochondrial-targeted TEMPO (mitoTEMPO, Sigma-Aldrich SML0737-25MG), mito-CoQH2 (Mitoquinol, Cayman Chemical Company CAY89950), and CoQ10 (Coenzyme Q10, Sigma-Aldrich C9538).

## Clonogenic survival assays and in vitro irradiation

It has already been shown that ferroptosis is highly dependent on the exact culture conditions, especially on the confluence of the cells in culture[63]. Therefore, we performed our experiments at approximately 50% confluence. In particular, in cell lines where LDHB silencing alone reduced cell number, more cells were seeded into the corresponding wells so that the number of cells per well was similar at the beginning of the treatment. Unless directly indicated otherwise in the figure, 24 h after siCTRL and siLDHB transfection, the following cell numbers were seeded per six-well: 1000 and 1500 cells for the A549 cell line, 500 cells for both conditions for the HT1080 cell line, and 500 and 1000 cells for the MSTO-211H cell line. Twenty-four hours later (48 h after transfection), mitoTEMPO was added as a pretreatment for 24 hours. After 24 hours (72 h after transfection), the corresponding plates were irradiated with an X-RAD 225 irradiator with 0.3 mm Cu filter (Precision X-Ray) at different doses of 0–6 Gy, and the control plates were mock-treated. After 9–10 days, colonies were fixed and stained with crystal violet (0.5% dissolved in 25% methanol). Images of the clones were then captured using a camera mounted in a Kaiser eVision high-frequency illuminated copy stand to avoid shadows. To determine the number of colonies per well, the images were then analyzed using Fiji software (Fiji, RRID:SCR_002285) as previously described[11]. The survival fraction was calculated using GraphPad Prism 9 after normalization to the untreated group. The survival curve was plotted using linear quadratic cell death as previously described[29].

## Lipid peroxidation analysis

Cells were harvested with TrypLE™ Select Enzyme (Thermo Fisher Scientific A1217702) and resuspended in 200 μL PBS containing 5 μM BODIPY™ 581/591 C11 (Invitrogen D3861) or 5 μM MitoPerOx (Abcam ab146820) for 30 min to measure total lipid peroxidation or mitochondrial lipid peroxidation. Lipid peroxidation of at least 10,000 cells was analyzed using a BD LSR-II flow cytometer with a 488-nm laser and FITC filter or a ZEISS_LSM 710 confocal microscope with a 63× oil immersion objective.

## Cell viability assay

Cell viability was measured using PrestoBlue™ Cell Viability Reagent (Thermo Fisher Scientific A13261) or Acid Phosphatase (APH) Assay. In detail, $1$–$4 \times 10^5$ cells per well were seeded in 96-well plates. The next day, cells were treated with ferrostatin-1 or TEMPO (Sigma-Aldrich 214000-1 G) or mitoTEMPO for 24 h, followed by RSL3 or ML162 for 48 h in 100 μL cell culture medium. For the PrestoBlue-based cell viability assay, 10 μL PrestoBlue™ Cell Viability Reagent was added in the culture medium. After incubation at 37 °C for 45 min, fluorescence was measured using a Varioskan™ LUX multimode microplate reader (Thermo Fisher Scientific) with a fluorescence excitation wavelength of 560 nm and an emission wavelength of 590 nm. The cell viability assay based on the APH assay was described in accordance with previous research[64]. The viability of the cells was determined in the experiments with PretoBlue, unless explicitly stated.

## Cell death assay

Cells were seeded in six-well plates and reached approximately 50% confluence on the day of treatment. After treatment with different regents, both floating cells and adherent cells were harvested and stained with 5 μg/ml PI (Sigma-Aldrich P486) immediately before flow cytometric analysis. At least 10,000 cells were analyzed using a BD LSR-II flow cytometer with a 488-nm laser and PI filter.

## Immunoblotting

Cell lysates were extracted in $1 \times$ RIPA lysis and extraction buffer (Sigma-Aldrich Chemie GmbH R0278) containing $2 \times$ Protease and Phosphatase Inhibitor Cocktail (Thermo Scientific 78444) for 20 min on ice. The lysate was purified by centrifugation at $14,000 \times g$ for 25 min

at 4 °C. Protein concentration was quantified using the BCA Protein Assay Kit (Thermo Fisher Scientific 23209). About 20–30 μg protein samples were resolved by SurePAGE Bis-Tris 10 × 8, 4–20% (Witec M00655) and then transferred using Trans-Blot® Turbo™ Mini Nitrocellulose Transfer Packs (Bio-Rad 1704158). Prior to staining with antibodies, membranes were blocked with TBS (LI-COR Biosciences 927-60001) for 2 h at room temperature. Subsequently, the membranes were incubated with the primary antibodies against LDHB (1:10,000, R&D Systems MAB9205-100), LDHA (1:1000, Cell Signaling Technology 2012S), GPX4 (1:1000, Abcam ab125066), DHODH (1:1000, Cell Signaling Technology 26381S), FSP1 (1:1000, Proteintech 20886-1-AP), SLC7A11 (1:1000, Cell Signaling Technology 12691S), beta-actin (1:5000, Cell Signaling Technology 3700S) overnight on a rotating wheel (3 rpm) at 4 °C. After washing three times with TBS wash buffer containing 0.2% TWEEN 20 (Sigma-Aldrich P1379), membranes were incubated for 45 min at room temperature with secondary antibodies anti-mouse IgG (1:5000; LI-COR Biosciences 926-68020) and anti-rabbit IgG (1:5000; LI-COR Biosciences 926-32211). Images were acquired and analyzed using the Odyssey Infrared Imaging System (Li-COR Biosciences).

## Immunofluorescence microscopy

Immunofluorescence was performed as previously described[11]. In brief, 20,000 cells were seeded in four-well chamber slides (Thermo Scientific Nunc 154526) for a 2-day culture. Cells were then fixed with 4% paraformaldehyde for 20 min at RT and permeabilized with 0.2% Triton X-100 for 15 min. Cells were stained with the TOM20 (1:100, Cell Signaling Technology, 42406S) and 4-hydroxynonenal (4-HNE) monoclonal antibody (1:100, Novus biological, NBP2-59353) overnight after cells were blocked with 1% BSA for 2 h at room temperature. Subsequently, cells were stained with F(ab')2-goat anti-rabbit IgG (H + L) cross-adsorbed secondary antibody, Alexa Fluor™ 546 (1:100, Thermo Fisher Scientific, A11071), or goat anti-mouse IgG (H + L) highly cross-adsorbed secondary antibody, Alexa Fluor™ 488 (1:100, Thermo Fisher Scientific, A11029). Images were acquired by ZEISS_LSM 710 confocal microscope and processed by Fiji.

## Immunohistochemistry

Immunohistochemistry was performed at room temperature using the fully automated BOND RX® staining system (Leica Biosystems) as previously described[65]. Samples were stained with appropriate antibodies against LDHB (1:5000, R&D Systems MAB9205-100), 4-HNE (1:200, Novus biological, NBP2-59353), GPX4 (1:800, Abcam ab125066), and caspase-3 (1:200, Cell Signaling, 9664 s). Images were acquired and processed using QuPath[66].

Immunohistochemistry was performed at room temperature using the fully automated BOND RX® staining system (Leica Biosystems) as previously described[65]. Samples were stained with appropriate antibodies against LDHB (1:5000, R&D Systems MAB9205-100) and 4-HNE (1:500, Abcam ab46545). Images were acquired and processed using QuPath[66].

## LDHB activity

LDHB activity was measured by the LDHB Activity Assay kit (Abcam ab140361) according to the manufacturer's instructions.

## NADH/NAD measurement

The NAD+ and NADH was measured using a NAD+/NADH assay kit (colorimetric) (BioVision K337-100) according to the manufacturer's protocol. In brief, $2 \times 10^5$ cells were harvested for extraction of total NAD+ and NADH (NADt) using the NADH/NAD extraction buffer. To measure NADH, the samples were heated at 60 °C for 30 min. The samples were mixed with 100 μL of the Reaction Mix or Background Control Mix. After incubating the plate at room temperature for 5 min, 10 μl NADH developer was added to each well. About 10 μl of Stop

Solution was added into each after 3 h incubation at room temperature. NADt and NADH were measured at OD450nm. The NAD + /NADH ratio was calculated as follows: NAD+/NADH ratio = (NADt-NADH)/NADH.

## Cellular ROS detection

The cellular ROS was detected using a Cellular Reactive Oxygen Species Detection Assay Kit (Abcam ab186029) according to the manufacturer's protocol. In brief, cells were harvested and washed twice with PBS, then stained with ROS Deep Red staining solution in a 37 °C/5% $CO_2$ incubator for 30 min. 100 μM TBHP (tert-butyl-hydroperoxide, Sigma-Aldrich 458139) treated samples were used as a positive control for the assay. The fluorescence was measured using a flow cytometer with Alexa Fluor 647 filter.

## Gene silencing by small interfering (siRNA) and short hairpin RNAs (shRNA)

Transient and stable gene silencing by siRNA and shRNA, respectively, was performed as previously described[11]. Briefly, cells were cultured in six-well plates until they reached 50–70%. Then pooled 10 nM universal scrambled negative control siRNA or LDHB human siRNA oligo duplex (3 unique 27mer siRNA duplexes, Cat. #SR320835; Origene) with Lipofectamine 2000 (Cat. #11668027; Invitrogen) was added to the P/S-free medium for 6 h of transfection. Cells were then changed to fresh growth medium until harvest.

Lentivirus was produced in 293T cells using Lipofectamine 2000 (9 μg packaging plasmid psPAX2 (Addgene 12260), 0.9 μg envelope plasmid pCAG-VSVG (Addgene 35616), 9 μg scramble control or shLDHB vectors (Origene TL311768) in 225 μL Opti-MEM (Thermo Fischer 31985062) mixed with 90 μL Opti-MEM containing 54 μL Lipofectamine 2000, 10-cm dish), the medium was changed after 18 h of incubation, then the virus was collected 24 and 48 h later. The lentiviruses were used to infect cells with 8 μg/mL polybrene for 72 h. Cells were then selected with 1–2 μg/mL puromycin (Sigma-Aldrich P8833-25MG) for 72 h. The inhibition of LDHB expression was verified by Western blot.

## CRISPR/Cas9 mediated KO

LentiCRISPR v2-sgNTC (non-target control), sgFSP1 (FSP1 knockout), sgDHODH-2, sgDHODH-4 (DHODH knockouts) were a gift from Boyi Gan (Addgene plasmid #125836; http://n2t.net/addgene:125836; RRID: Addgene_125836; Addgene plasmid #186026; http://n2t.net/addgene:186026; RRID: Addgene_186026, Addgene plasmid #186023; ttp://n2t.net/addgene:186023; RRID: Addgene_186023, Addgene plasmid #186022; http://n2t.net/addgene:186022; RRID: Addgene_186022. The non-target control and GPX4 knockout plasmids were constructed in the pLentiCRISPRv2 vector by Genescript. The target sequences of GPX4-1 and GPX4-3 are CACGCCCGATACGCTGAGTG and GAATTT-GACGTTGTAGCCCG, respectively. The lentiviruses were produced as described above. The virus-infected cells were selected with 1–2 μg/mL puromycin.

## Overexpression cell line generation

CCSB-Broad LentiORF-LDHB Clone and empty vector were purchased from Horizon Discovery. The lentiviruses were produced as described above. The virus-infected cells were selected with 20 μg/mL blasticidin.

## Mitochondrial isolation and preparation of cellular subfractions

The mitochondrial isolation and preparation of subfractions were described previously[26]. In brief, mitochondria were isolated from $50 \times 10^6$ cells using a Mitochondria/Cytosol fractionation kit (Abcam ab65320). For the preparation of mitochondrial subfractions, 1 mg of isolated mitochondria was suspended with 40 μl T10E20 buffer (10 mM Tris-HCL, 1 mM EDTA, pH 7.6) and then incubated with digitonin (0.1 mg digitonin/mg of mitochondria protein) for 10 min on ice. Then, 3 volumes of 250 mM sucrose were added to the mitochondrial suspension. The inner mitochondrial membrane and mitochondrial matrix were precipitated, and the outer membrane and intermembrane components were in the supernatant after centrifugation at 10,000×$g$ for 15 min at 4 °C. For Western blot analysis, the same amounts of inner and outer mitochondrial membrane were mixed with 1 × sample loading buffer. Samples are then heated at 70 °C for 10 min.

## CoQ and CoQH2 analysis

About $3 \times 10^6$ cells were harvested using TrypLE™ Select Enzyme (Thermo Fisher Scientific A1217702). Wash the collected cells twice by resuspending them in "D-PBS without Mg, Ca" (Thermo Fisher Scientific 14190136) and spin down. Remove the D-PBS buffer and save the cell pellet. Freeze the pellet and store it at −80 °C. The whole sample was extracted with 2-propanol using sonication. The extract was centrifuged and injected directly. The clear solutions were analysed on an Agilent 1290 HPLC system with a binary pump, multisampler, and column thermostat with a Zorbax Eclipse C-18, 3.0 × 50 mm, 1.8 μm column using methanol/2-propanol. HPLC was coupled to an Agilent 6495 Triplequad mass spectrometer (Agilent Technologies, Santa Clara, USA) with an electrospray ionization source. Analysis was performed with Multiple Reaction Monitoring in positive mode, with at least two mass transitions for each compound. The dynamic range was determined prior to analysis. Based on these data, the limits of quantification and coefficients of variation were determined for the different lipid classes. The limits of quantification are in the lower picogram range depending on the analyte. The average coefficient of variation for a complete set of analytes is <15%.

## Transmission electron microscopy

Cells were seeded in six-well plates at 50% confluence. After removal of the cell culture media, the cells were submerged with a fixative, which was prepared as follows: 2.5% glutaraldehyde (Agar Scientific, Stansted, Essex, UK) in 0.15 M HEPES (Fluka, Buchs, Switzerland) with an osmolarity of 670 mOsm and adjusted to a pH of 7.35. The cells remained in the fixative at 4 °C for at least 24 h before being further processed. They were then washed with 0.15 M HEPES three times for 5 min, postfixed with 1% OsO4 (EMS, Hatfield, USA) in 0.1 M Na-cacodylate-buffer (Merck, Darmstadt, Germany) at 4 °C for 1 h. Thereafter, cells were washed in 0.1 M Na-cacodylate buffer three times for 5 min and dehydrated in 70, 80, and 96% ethanol (Grogg Chemie, Bern, Switzerland) for 15 min each at room temperature. Subsequently, cells were immersed in 100% ethanol (Merck, Darmstadt, Germany) three times for 10 min, and finally in ethanol-Epon (1:1) overnight at room temperature. The next day, cells were embedded in Epon (Sigma-Aldrich, Buchs, Switzerland) and left to harden at 60 °C for 5 days. Sections were produced with an ultramicrotome UC6 (Leica Microsystems, Vienna, Austria), with a thickness of 70–80 nm. The sections, mounted on 200 mesh copper grids, were stained with uranyless (EMS, Hatfield, USA) and lead citrate with an ultrastainer (Leica Microsystems, Vienna, Austria). Sections were then examined with a transmission electron microscope (Tecnai Spirit, FEI, Brno, Czech Republic) equipped with a digital camera (Veleta, Olympus, Soft Imaging System, Münster, Germany).

## Xenograft model

The mouse experiments were performed in accordance with animal welfare guidelines and protocols approved by the Institutional Animal Care and Ethical Committee (license number 34192). All animals were housed on a 12:12 h light–dark cycle. The temperature in both housing rooms ranged from 21 to 23 °C and relative humidity was between 40–56%. The animals were fed irradiated food provided ad libitum (Granovit AG, 343200PXV20), and water was also available ad libitum.

All animal studies included both male and female animals aged 6 to 10 weeks. Animals of both sexes, aged 6 to 10 weeks, were used in all animal studies. For RSL3 treatment, $1 \times 10^6$ A549 cells were suspended in 100 μl PBS and growth factor-reduced Matrigel (1:1) (Corning 356231) and injected subcutaneously (left and right flank) into NSG mice. After 3 days, 100 mg/kg RSL3 or the same volume of PEG-300 was injected at the site where the cancer cells were injected twice per week for 2 weeks. Alternatively, $1 \times 10^6$ A549 and HT1080 sgNTC shCTRL, sgNTC shLDHB, sgGPX4 shCTRL, and sgGPX4 shLDHB cell suspension were injected subcutaneously into NSG animals. The next day, the mice were treated with 10 mg/kg liproxstatin-1 daily by i.p. injection. For local irradiation, $1 \times 10^6$ cancer cells suspended in 100 μl PBS and growth factor-reduced Matrigel (1:1) were injected subcutaneously (left and right flank) into RAG mice. When tumor volume reached 40–100 mm³, the mice were randomly assigned to different treatment groups and then were irradiated locally using an X-RAD SmART irradiator (Precision X-Ray) with a single dose of 10 Gy. The tumor size was measured twice per week, and the tumor volume was calculated according to the flowing equation: $volume = length \times width^2/2$. The maximum size permitted for subcutaneous tumors, as specified by the local ethics committee, was 1000 mm³, and this limit was not exceeded in this study. Experiments involving orthotopic tumors were terminated after approximately 35 days or when the lung volume reduction reached a maximum of 200 mm³, in accordance with the guidelines set by the local ethics committee.

### Genetically engineered mouse model

The genetically engineered mouse model was established as described previously[11]. In brief, $1 \times 10^9$ VG AAV-Cre virus in a total volume of 50 μl was introduced to Ldhb[+/+]; K-ras[LSL-G12D/+]; p53[fl/fl] and Ldhb[-/-]; K-ras[LSL-G12D/+]; p53[fl/fl] mice age at 8 weeks of age after anesthesia. Two weeks later, the development of the lung tumor was monitored once a week with a microCT (X-RAD SmART, Precision X-Ray) using PilotRad software (version 1.18.5.2) with the following setup for scan: Mouse Soft Tissue High Dose 0.1 mm voxels with 2 mm AI filter and the microCT DICOM data were exported with the following setup: Slope:2617, Intercept: -1002. The microCT data were imported into the treatment planning software (SmART-ATP version 2.0.20191017) to create the treatment plan. The treatment plan was then imported into the PilotRad software for treatment. Lung tumors with a volume of approximately 1 mm³ were irradiated locally with 24 Gy over the course of 3 days using an X-RAD SmART irradiator at 225 kV X-RAY (0.3-mm Cu filter) with a 3 mm collimator. Subsequently, mice were scanned with microCT once a week to assess tumor development and sacrificed after 5–6 weeks of treatment. The microCT images were processed and analyzed with 3D Slicer version 4.13 according to a previously published protocol[67,68].

### Public data source and analysis

Data for correlation analysis of LDHB and GPX4 were acquired and analyzed using cBioPortal[69]. The single-cell sequencing data were collected and analyzed in Single Cell Expression Atlas[70]. Overall survival data and analysis were obtained and analyzed using the Kaplan–Meier plotter (Kaplan–Meier plotter [Lung] (kmplot.com)).

### Statistical analysis

Statistical analysis was performed using GraphPad Prism 9. Results were collected from at least three independent replicates or eight tumors for in vitro and in vivo experiments. Error bars represent mean ± standard deviation (SD) or mean ± standard error of the mean (SEM). Ordinary one-way ANOVA and two-tailed unpaired Student's $t$-tests were performed as described in the figure legends. The $p$ values <0.05 were considered significant. In all analyses, the significance level is reported as follows: $*P < 0.05$, $**P < 0.01$, $***P < 0.001$, and $****P < 0.0001$.

### Reporting summary

Further information on research design is available in the Nature Portfolio Reporting Summary linked to this article.

### Data availability

All data generated or analyzed during this study are included in this published article and its supplementary information files. Source data are provided with this paper.

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

## Acknowledgements

Flow cytometry experiments were performed with the support of the FACS Lab at the University of Bern, Switzerland. Microscopy acquisition and analysis were performed with the support of the Live Cell Imaging Core Facility of the Department of Clinical Research, coordinated by the Microscopy Imaging Center at the University of Bern, Switzerland. Electron microscopy sample preparation and imaging were performed with devices supported by the Microscopy Imaging Center (MIC) of the University of Bern, Switzerland. The combination of the KP and LDHB knockout animal models was generated in collaboration with Dr. Urban Deutsch at the Central Experimental Mouse Facility Bühlplatz (ZEMB) at the Theodor–Kocher-Institute of the University of Bern. Animal experiments were performed in collaboration with the Experimental Animal Center (EAC) of the Medical Faculty of the University of Berne. We thank Prof. Boyi Gan for reading the manuscript and providing comments. This work was supported by the Swiss Cancer Research (KFS-5405-08-2021) and the Swiss National Science Foundation (310030_2127661) to T.M.M. The research contribution of RWP was funded by the Swiss National Science Foundation (310030_192648 and 320030-231251). The research contribution of W.W. was funded by the Hunan Provincial Natural Science Foundation (2021JJ70103) and the Health Research Project of Hunan Provincial Health Commission (C2019074). The funding bodies were not involved in the design of the study and collection, analysis, and interpretation of data and in writing the manuscript.

## Author contributions

Conception and design—H.D., W.W., R.-W.P., P.D., and T.M.M.; Data acquisition—H.D., L.Z., H.G., Y.G., Y.F., Y.L., M.M., T.L., M.M., and J.O.; Data interpretation and analysis—H.D., L.Z., H.G., Y.G., Y.F., Y.L., M.M., T.L., M.S., W.W., R.W.P., P.D., and T.M.M.; Drafting of manuscript—H.D., R.W.P., P.D., and T.M.M.; Accountability for all aspects of work—H.D., W.W., R.-W.P., P.D., and T.M.M. All authors read and approved the final manuscript.

## Competing interests

The authors declare no competing interests.

## Additional information

[1]2nd Department of Thoracic Surgery, Hunan Cancer Hospital and The Affiliated Cancer Hospital of Xiangya School of Medicine, Central South University, Changsha, Hunan 410013, China. [2]Hunan Clinical Medical Research Center of Accurate Diagnosis and Treatment for esophageal carcinoma, Hunan Cancer Hospital and The Affiliated Cancer Hospital of Xiangya School of Medicine, Central South University, Changsha, Hunan 410013, China. [3]Department of General Thoracic Surgery, Inselspital, Bern University Hospital, Bern, Switzerland. [4]Department for BioMedical Research, University of Bern, Bern, Switzerland. [5]Graduate School of Cellular and Biomedical Sciences, University of Bern, Bern, Switzerland. [6]Institute of Clinical Chemistry, Inselspital, Bern University Hospital, Bern, Switzerland. [7]Institute of Tissue Medicine and Pathology, ITMP, University of Bern, Bern, Switzerland. [8]Department of Radiation Oncology, Inselspital, Bern University Hospital, Bern, Switzerland. ✉e-mail: renwang.peng@insel.ch; patrick.dorn@insel.ch; thomas.marti@insel.ch

