## [Transparent Peer Review file · Nature Communications]

Ubiquinol-mediated suppression of mitochondria-associated ferroptosis is a targetable function of lactate dehydrogenase B in cancer

Corresponding Author: Dr Thomas Marti

Version 0:

Reviewer comments:

Reviewer #1

(Remarks to the Author)

The authors have made a concerted effort to adapt the manuscript to make it more balanced and attempted to action the majority of the suggestions. For my part I believe the manuscript is now suitable for publication in Nature Communications.

Do note that some labelling of the main Figures has been lost during revision i.e. Figure 2e where the PANC-1 and MSTO-211H do not have the siRNA labelled (siCTRL and siLDHB), so the authors should make sure all Figures and Legends are correct.

For the MitoEbselen-2 data, there is no need to add it to the Supplemental Figures as there is already a lot of supplemental data.

Reviewer #2

(Remarks to the Author)

The authors have addressed most of my concerns.

Reviewer #4

(Remarks to the Author)

The authors have submitted a revised version of their manuscript, "Ubiquinol-mediated suppression of mitochondria-associated ferroptosis is a novel and targetable function of lactate dehydrogenase B in cancer", in which they describe that LDHB-mediated reduction of the CoQ pool contributes to mitochondrial membrane lipid homeostasis and the mitigation of ferroptosis. The authors have performed a substantial and commendable amount of additional experimental work to largely address the concerns of the reviewers and to substantiate the underlying conclusion that LDHB loss propagates mitochondrial lipid peroxidation and sensitizes cancer cells to inducers of ferroptosis. The mechanistic gaps regarding LDHB function and the metabolic consequences of its silencing raised by reviewer 3 remain, though the authors have done an adequate job of providing hypothetical explanations for their findings in the discussion. I have some additional concerns listed below, though most can be addressed without additional experiments.

Major Comments:

Could the authors provide quantitative analysis of their TEM images to confirm their conclusions of changes in mitochondrial morphology in response to LDHB silencing?

If the specific argument for this study revolves around mitochondrial lipid peroxidation, why were mitochondrial specific ROS dyes (MitoPY1 [H₂O₂], MitoSox [•O₂-], or even OH850 [•OH]) not used to assess if there was a change in total mitochondrial

ROS in association with LDHB silencing?

Does LDHB silencing similarly sensitize cancer cells to cysteine starvation? The authors title the section related to Figures 2 and S2 "LDHB suppresses mitochondria-associated ferroptosis in cancer cells". As the authors cite themselves, there is a clear link between mitochondrial oxidative metabolism and cysteine-starvation induced ferroptosis in cancer cells (Gao, M et al. [2015], PMID: 26166707; Gao, M et al. [2019], PMID: 30582246; Homma, T et al., PMID: 33493440; Lee, H et al., PMID: 32029897; Ward NP et al., PMID: 38762605). There are defined distinctions between different inducers of ferroptosis (Magtanong, L et al., PMID: 35809566) and the authors should be more specific in the interpretation of their findings.

How does LDHB restrict GPX4 expression? This is an interesting finding, can the authors provide discussion on a potential mechanism that could tie the redox function of LDHB to the control of GPX4 expression?

Minor Comments:

Does Fer-1 rescue viability in untreated siLDHB cells?

The quality or resolution of Figure 3c makes it impossible to ascertain the morphological characteristics described by the authors.

Legend color and sgGPX4+shLDHB+Lip1 plots do not match for Figures 3l and 3o.

Why are panels from Figures 4 and S4 referenced out of order in the manuscript text relative to their organization in the figures themselves?

Referees' comments:

Title: Ubiquinol-mediated suppression of mitochondria-associated ferroptosis is a novel and targetable function of lactate dehydrogenase B in cancer

REVIEWERS' COMMENTS

Reviewer #1 (Remarks to the Author):

The authors have made a concerted effort to adapt the manuscript to make it more balanced and attempted to action the majority of the suggestions. For my part I believe the manuscript is now suitable for publication in Nature Communications.

Thank you for your thoughtful feedback and support; we greatly appreciate your recognition of our efforts and are delighted that you find the manuscript suitable for publication.

Do note that some labelling of the main Figures has been lost during revision i.e. Figure 2e where the PANC-1 and MSTO-211H do not have the siRNA labelled (siCTRL and siLDHB), so the authors should make sure all Figures and Legends are correct.

Thank you for pointing this out; we have corrected the labeling in Figure 2e and ensured all Figures and Legends are accurate.

For the MitoEbselen-2 data, there is no need to add it to the Supplemental Figures as there is already a lot of supplemental data.

Thank you for agreeing that the MitoEbselen-2 data does not need to be added to the Supplemental Figures.

Reviewer #2 (Remarks to the Author):

The authors have addressed most of my concerns.

Thank you for your kind feedback and for acknowledging our efforts to address your concerns.

Reviewer #3, Withdrawn

Reviewer #4 (Remarks to the Author):

The authors have submitted a revised version of their manuscript, "Ubiquinol-mediated suppression of mitochondria-associated ferroptosis is a novel and targetable function of lactate dehydrogenase B in cancer", in which they describe that LDHB-mediated reduction of the CoQ pool contributes to mitochondrial membrane lipid homeostasis and the mitigation of ferroptosis. The authors have performed a substantial and commendable amount of additional experimental work to largely address the concerns of the reviewers and to substantiate the underlying conclusion that LDHB loss propagates mitochondrial lipid peroxidation and sensitizes cancer cells to inducers of ferroptosis. The mechanistic gaps regarding LDHB function and the metabolic consequences of its silencing raised by reviewer 3 remain, though the authors have done an adequate job of providing hypothetical explanations for their findings in the discussion. I have some additional concerns listed below, though most can be addressed without additional experiments.

Thank you for your detailed and thoughtful feedback, as well as for recognizing our additional experimental efforts.

Major Comments:

Could the authors provide quantitative analysis of their TEM images to confirm their conclusions of changes in mitochondrial morphology in response to LDHB silencing?

In the original paper by Dixon et al. that introduced the term ferroptosis (1), TEM images were described qualitatively rather than through quantitative analysis of specific features. Subsequent studies (2-4) similarly focused on qualitative evaluations of mitochondrial morphology following ferroptosis induction. In line with this precedent, we also refrained from performing quantitative analysis of our TEM images due to the limited number of cells available for examination. Instead, we used flow cytometry to analyze mitochondrial lipid peroxidation, enabling quantitative assessment of 10,000 cells and reducing potential bias. However, if the editor insists, we would agree to perform a quantitative analysis of our TEM images.

If the specific argument for this study revolves around mitochondrial lipid peroxidation, why were mitochondrial specific ROS dyes (MitoPY1 [H₂O₂], MitoSox [•O₂-], or even OH850 [•OH]) not used to assess if there was a change in total mitochondrial ROS in association with LDHB silencing?

In this study, we reported the following in our results section: "However, LDHB silencing did not result to a dramatic increase in cellular ROS levels. In detail, cellular ROS levels were not altered in A549 and HT1080 cells. Compared to ROS induction by treatment with tert-butyl hydroperoxide (TBHP), an exogenous inducer of oxidative stress 37, ROS levels were only slightly increased in PANC-1 cells and actually decreased MSTO-211H cells (Fig. S1q-r)". This is in agreement with our findings published earlier this year in the study by Ge et al. (5). We reported in detail: "Since a reduced GSH/GSSG ratio is associated with oxidative stress, we examined the changes in cellular ROS and mitochondrial ROS after short-term and long-term LDHB silencing. Surprisingly, neither cellular ROS nor mitochondrial ROS increased significantly after short-term LDHB silencing in both A549 (Fig. 2C) and PF139 cells (Fig. 2D). Also, treatment of A549 and PF139 cells with tert-butyl hydroperoxide (TBHP), a chemical compound commonly used to induce oxidative stress in biological systems [22], increased cellular and mitochondrial ROS levels, which was not further enhanced by additional silencing of LDHB (Fig. 2C and D).

Consequently, we have updated our results section accordingly (the added text is highlighted in yellow): “However, LDHB silencing did not result to a dramatic increase in cellular ROS levels. In detail, 72 hours after LDHB silencing, cellular ROS levels were not altered in A549 and HT1080 cells. Compared to ROS induction by treatment with tert-butyl hydroperoxide (TBHP), an exogenous inducer of oxidative stress (6), ROS levels were only slightly increased in PANC-1 cells and actually decreased in MSTO-211H cells (Fig. S1q-r). This is consistent with our recently published findings, which showed that total and mitochondrial ROS levels in A549 cells were not significantly increased 48 hours after LDHB silencing (5).

Does LDHB silencing similarly sensitize cancer cells to cysteine starvation? The authors title the section related to Figures 2 and S2 “LDHB suppresses mitochondria-associated ferroptosis in cancer cells”. As the authors cite themselves, there is a clear link between mitochondrial oxidative metabolism and cysteine-starvation induced ferroptosis in cancer cells (Gao, M et al. [2015], PMID: 26166707; Gao, M et al. [2019], PMID: 30582246; Homma, T et al., PMID: 33493440; Lee, H et al., PMID: 32029897; Ward NP et al., PMID: 38762605). There are defined distinctions between different inducers of ferroptosis (Magtanong, L et al., PMID: 35809566) and the authors should be more specific in the interpretation of their findings.

We would like to thank the reviewer for this constructive comment. Consequently, we added the following paragraph to our discussion section: “). Finally, defined distinctions exist between different ferroptosis inducers (7). Thus, it would be interesting to test how LDHB affects the response of cancer cells to ferroptosis induction by cysteine-starvation, which was shown to also affect mitochondrial oxidative metabolism (8).

How does LDHB restrict GPX4 expression? This is an interesting finding, can the authors provide discussion on a potential mechanism that could tie the redox function of LDHB to the control of GPX4 expression?

We agree with the reviewer that this indeed is an interesting finding. We are currently working on elucidating the underlying molecular mechanism. Consequently, we adapted the discussion section (the added text is highlighted in yellow): However, further experiments are needed to determine the exact relative contributions and to uncover the molecular mechanism behind the observed increase in total cellular GPX4 protein levels following LDHB silencing (Fig. 3a and Fig. S3a-b).

Minor Comments:

Does Fer-1 rescue viability in untreated siLDHB cells?

No. We previously reported that silencing LDHB significantly reduces mitochondrial metabolism. While FER1 can rescue lipid peroxidation, it does not restore other mitochondrial functions, such as nucleotide metabolism. Currently, we have two publications under review in which we describe the role of LDHB-mediated nucleotide metabolism in the DNA damage response, particularly in the context of DNA damage induced by chemotherapy or ionizing radiation treatment.

This was already described in our discussion section of our original submission, i.e., “However, LDHB silencing affects more than just lipid peroxidation, as the reduction in colony formation was only partially rescued by FER1 and mitoTEMPO treatment (Fig. 1h-i). This is consistent with our previous data showing that LDHB silencing broadly impacts mitochondrial metabolism 11”.

The quality or resolution of Figure 3c makes it impossible to ascertain the morphological characteristics described by the authors.

We appreciate the reviewer's constructive comment. In response, we have replaced the images with higher-resolution versions.

Legend color and sgGPX4+shLDHB+Lip1 plots do not match for Figures 3l and 3o.

We would like to thank the reviewer for this constructive comment. We have updated the figures accordingly.

Why are panels from Figures 4 and S4 referenced out of order in the manuscript text relative to their organization in the figures themselves?

Figure 4 and Figure S4 contain 26 and 30 individual panels, respectively. We arranged the panels to maximize the information presented while maintaining a clear and simple layout. Rearranging Figure 4 as suggested would require transferring some of the information to Figure S4. To preserve the information density of Figure 4, we prefer to keep its current layout. However, if the editor insists, we are willing to rearrange the panels in both figures and update the manuscript and figure legends accordingly.

Finally, we made several minor edits in the revised version of the manuscript.

-First, our manuscript by Deng et al. focuses on the role of LDHB in suppressing CoQH2-mediated, mitochondria-associated ferroptosis. As also mentioned in the discussion section of this manuscript, our manuscript by Zhao et al. focuses on the role of LDHB on regulating SLC7A11-mediated suppression of ferroptosis of the plasma membrane. The manuscript by Zhao et al. has been published during the revision process. Thus we updated the corresponding sentence in the introduction section: “Further, we described that LDHB protects specifically KRAS-mutant lung cancer cells from ferroptosis, mainly of the plasma membrane, by maintaining SLC7A11-mediated

glutathione metabolism 24" and also updated the corresponding reference #24 with the permanent DOI.

REFERENCES:

1. Scott *et al.*, Ferroptosis: An Iron-Dependent Form of Nonapoptotic Cell Death. *Cell* **149**, 1060-1072 (2012).
2. G. Lei *et al.*, The role of ferroptosis in ionizing radiation-induced cell death and tumor suppression. *Cell Research* **30**, 146-162 (2020).
3. N. Yagoda *et al.*, RAS–RAF–MEK-dependent oxidative cell death involving voltage-dependent anion channels. *Nature* **447**, 865-869 (2007).
4. H. Lee *et al.*, Energy-stress-mediated AMPK activation inhibits ferroptosis. *Nature cell biology* **22**, 225-234 (2020).
5. H. Ge *et al.*, Inhibition of LDHB suppresses the metastatic potential of lung cancer by reducing mitochondrial GSH catabolism. *Cancer letters* **611**, 217353 (2025).
6. O. Kučera *et al.*, The effect of tert-butyl hydroperoxide-induced oxidative stress on lean and steatotic rat hepatocytes in vitro. *Oxid Med Cell Longev* **2014**, 752506 (2014).
7. L. Magtanong *et al.*, Context-dependent regulation of ferroptosis sensitivity. *Cell chemical biology* **29**, 1409-1418.e1406 (2022).
8. M. Gao *et al.*, Role of Mitochondria in Ferroptosis. *Molecular Cell* **73**, 354-363.e353 (2019).